# Advanced Nanostructured Materials for Electrocatalysis in Lithium–Sulfur Batteries

**DOI:** 10.3390/nano12234341

**Published:** 2022-12-06

**Authors:** Zihui Song, Wanyuan Jiang, Xigao Jian, Fangyuan Hu

**Affiliations:** 1School of Materials Science and Engineering, State Key Laboratory of Fine Chemicals, Frontiers Science Center for Smart Materials Oriented Chemical Engineering, Technology Innovation Center of High Performance Resin Materials (Liaoning Province), Key Laboratory of Energy Materials and Devices (Liaoning Province), Dalian University of Technology, Dalian 116024, China; 2State Key Laboratory of Fine Chemicals, Frontiers Science Center for Smart Materials Oriented Chemical Engineering, School of Chemical Engineering, Technology Innovation Center of High Performance Resin Materials (Liaoning Province), Key Laboratory of Energy Materials and Devices (Liaoning Province), Dalian University of Technology, Dalian 116024, China

**Keywords:** lithium–sulfur batteries, nanostructure, electrocatalysis, shuttle effect, redox reaction kinetics

## Abstract

Lithium–sulfur (Li-S) batteries are considered as among the most promising electrochemical energy storage devices due to their high theoretical energy density and low cost. However, the inherently complex electrochemical mechanism in Li-S batteries leads to problems such as slow internal reaction kinetics and a severe shuttle effect, which seriously affect the practical application of batteries. Therefore, accelerating the internal electrochemical reactions of Li-S batteries is the key to realize their large-scale applications. This article reviews significant efforts to address the above problems, mainly the catalysis of electrochemical reactions by specific nanostructured materials. Through the rational design of homogeneous and heterogeneous catalysts (including but not limited to strategies such as single atoms, heterostructures, metal compounds, and small-molecule solvents), the chemical reactivity of Li-S batteries has been effectively improved. Here, the application of nanomaterials in the field of electrocatalysis for Li-S batteries is introduced in detail, and the advancement of nanostructures in Li-S batteries is emphasized.

## 1. Introduction

With the development of society and economy, traditional fossil energy such as coal and oil are continuously consumed, and problems such as environmental pollution and ecological damage have become more serious. Development of green and renewable new energy has become an important task at present [1]. Most common renewable types of energy (solar, wind, hydro and tidal energy, etc.) have uncontrollable shortcomings and cannot be used conveniently. The electrochemical energy storage system can realize the mutual conversion of chemical energy and electrical energy, so as to realize the transformation, storage and utilization of energy, which has become the focus of research [2,3]. Among them, lithium-ion (Li-ion) batteries were first commercialized by Sony in 1991, and are now widely used in electric vehicles, portable electronic devices, energy storage and other fields [4]. However, due to the limitations of the working principle and the materials themselves, the performance of Li-ion batteries has so far been close to the theoretical value. Therefore, it is necessary to develop next-generation novel electrochemical energy storage devices to meet the growing energy demand [5,6].

Lithium–sulfur (Li-S) batteries have attracted extensive attention due to their excellent theoretical capacity (1675 mAh g^−1^) and energy density (2600 Wh kg^−1^), as well as the environmental friendliness and cost-effectiveness of elemental sulfur [7,8,9]. A typical Li-S battery that provides high capacity through multi-electron exchange reactions usually uses metallic lithium as the anode and sulfur-containing material as the cathode, and its electrolyte is mostly an organic liquid electrolyte [10,11]. According to the different sulfur forms in the cathode, the reaction mechanism of Li-S batteries is usually divided into two categories: the first is the “(quasi) solid phase” transformation mechanism, which mostly uses atomic/small-molecule sulfur cathodes represented by sulfurized polyacrylonitrile, and the electrolyte are mainly esters [12,13]. Sulfur is confined in the form of atom/small-molecule clusters or fixed by chemical bonds in porous carbon or polymer frameworks [14]. The charge–discharge process is manifested as a solid-phase transition between sulfur and lithium sulfide (Li_2_S), and the discharge curve presents a single platform [15,16] (Figure 1a). The second type of reaction process is the “solid–liquid–solid” transformation mechanism, which is more common [17,18,19]. Most composite materials containing elemental sulfur are used as the cathode electrode, and the ether solvent with good solubility for the intermediate-chain lithium polysulfide product (Li_2_S*_x_*, x = 4~8) is used as the electrolyte [20,21,22,23]. During the discharge process, the solid elemental sulfur is gradually reduced to Li_2_S*_x_*, which is dissolved in the electrolyte, and the active material is separated from the conductive substrate. After that, Li_2_S*_x_* is further reduced to form Li_2_S with low solubility, which is redeposited on the conductive substrate [24,25,26]. The specific reaction process can be represented by the following chemical equation:S_8_ (solid) + 2Li^+^ + 2e^−^ → Li_2_S_8_ (liquid)(1)
3Li_2_S_8_ (liquid) + 2Li^+^ + 2e^−^ → 4Li_2_S_6_ (liquid)(2)
Li_2_S_8_ (liquid) + 2Li^+^ + 2e^−^ → 2Li_2_S_4_ (liquid)(3)
2Li_2_S_6_ (liquid) + 2Li^+^ + 2e^−^ → 3Li_2_S_4_ (liquid)(4)
Li_2_S_4_ (liquid) + 2Li^+^ + 2e^−^ → 2Li_2_S_2_ (solid)(5)
Li_2_S_2_ (solid) + 2Li^+^ + 2e^−^ → 2Li_2_S (solid)(6)

The process exhibits a typical “solid–liquid–solid” multi-phase transformation mechanism, and the discharge curve presents two plateaus, corresponding to the solid–liquid transition and the liquid–solid transition, respectively [27,28] (Figure 1a). In particular, the first type of mechanism is highly dependent on the dispersion state of atomic/small-molecule sulfur, which greatly limits the sulfur content of the cathode [29]. Therefore, most of the research focuses on the second type of mechanism, and the complex multi-step electrochemical reaction process greatly limits the development of Li-S batteries [30,31].

The sluggish redox reaction kinetics causes a large amount of polysulfides dissolved in the electrolyte to fail to react rapidly and to diffuse to the anode under the influence of electric field force and concentration gradient, resulting in irreversible loss of active materials and consumption of a large amount of electrolyte [32,33]. The process, known as the shuttle effect, is the biggest problem that limits the performance improvement of Li-S batteries [34,35,36]. Meanwhile, the drastic change in cathode volume and the infinite growth of anode dendrites during the charge–discharge process further degrade the performance of Li-S batteries in practical applications [37,38]. In order to solve the above problems, strategies such as physical confinement and chemical adsorption have been proposed successively [39,40,41,42]. These strategies mainly include constructing cathodes with novel nanostructures to confine polysulfides [40]; modifying separators to suppress the shuttle effect [37]; introduction of additional interlayer materials with special properties to anchor polysulfides [43,44]; rationally designing novel electrolytes to confine polysulfide shuttles [45,46]; and constructing robust artificial solid electrolyte interfaces on Li anodes to suppress lithium dendrites [47]. Although the above strategies alleviate the existing problems of Li-S batteries to a certain extent through adsorption, the slow redox kinetics and the severe shuttle effect still cannot be completely solved, which inevitably leads to low-rate performance and a short cycle life [48,49,50]. Therefore, in addition to rationally designing the internal components of the battery, exploring how to accelerate its redox reaction kinetics is a feasible way to solve the existing problems of Li-S batteries [49].

As a fundamental concept in chemistry, catalysis generally accelerates the rate of a reaction by reducing the activation energy (*E*_a_) during the reaction [51,52]. In the field of electrochemical energy storage, electrocatalysis has received extensive attention with the continuous development of electrochemical applications such as fuel batteries and water electrolysis [53,54,55]. Inspired by the above applications, it is an effective solution to reasonably introduce electrocatalysts into Li-S batteries to accelerate the overall electrochemical reaction process rate and achieve high-speed conversion of polysulfides [56]. In 2015, the shuttle effect in Li-S batteries was shown for the first time to be controlled by electrocatalysis [57]. After that, heterogeneous catalysts including metal compounds [40,58,59,60], heterostructures [41,61], and single-atom catalysts [62,63,64,65] have been successively designed and introduced into Li-S battery systems. Subsequently, the concept of homogeneous catalysis in Li-S battery systems was first proposed on this basis [66]. With the deepening of research, researchers are increasingly aware of the important role of rationally designed electrocatalysts for Li-S batteries in avoiding the shuttle effect and accelerating the redox kinetics. So far, the concept of “Li-S battery electrocatalysis” has received more and more attention, and related research is also developing rapidly [58,67].

For efficient Li-S battery electrocatalysts (especially heterogeneous catalysts), in addition to the material properties and catalytic mechanism, reasonable nanostructure and molecular structure are also important parameters that determine the catalytic performance [68]. However, so far, few reviews have explored the effect of nanostructure and molecular structure on the catalytic performance of electrocatalysts for Li-S batteries. Here, we present a detailed review of electrocatalytic systems in Li-S batteries, mainly covering heterogeneous electrocatalytic systems and homogeneous electrocatalytic systems. Combined with the current research status, the design ideas of Li-S electrocatalysts are discussed in detail, and the key roles of various nanostructures and molecular structures in the design of electrocatalysts are emphasized. The nanostructures and functional groups of the two types of electrocatalysts are systematically summarized separately. The relationship between the amount of non-homogeneous electrocatalyst, the specific surface of the material structure and the electrochemical performance is also discussed. Finally, future design ideas are proposed based on the existing achievements, current challenges, and potential opportunities to re-examine the important development of nanostructures and molecular structures in Li-S battery electrocatalysts. In particular, a generic scheme comparing heterogeneous and homogeneous electrocatalyst mechanisms is reported in order to better explain the two different processes/mechanisms. All in all, trying to interpret the electrocatalysis of Li-S batteries from the chemical perspective of structure-determining properties will open up new ideas for the practical large-scale application of Li-S batteries.

## 2. Heterogeneous Catalysts

In chemical concepts, when the catalyst is in a different phase with the reactants, it is called a heterogeneous catalyst. For Li-S batteries, heterogeneous catalysts generally exist in the reaction system in the form of solid-state catalysts, which are insoluble in the electrolyte during the reaction process. Therefore, the heterogeneous catalysts in Li-S batteries are generally compounded on the cathode, separator, or in the form of interlayers (Figure 2). Heterogeneous catalysts mostly accelerate the chemical reaction rate of Li-S batteries by adsorbing dissolved polysulfides and promoting the transfer of electrons/ions to achieve rapid conversion between sulfur-containing species [69,70]. According to chemical types and nanostructures, this section divides heterogeneous catalysts into metal compounds, heterostructures and single-atom catalysis, and introduces the three types of heterogeneous catalysts in detail.

### 2.1. Metal Compounds

Metal materials have become among the most used electrocatalysts in industry due to their high electrical conductivity and good coordination. Traditional noble metal materials (Pt, Au, etc.) are widely used as catalysts in various electrochemical processes such as electroreduction and water electrolysis [71,72,73]. Inspired by this, traditional electrocatalyst Pt nanoparticles (NPs) [57] are loaded on graphene as the cathode host of Li-S batteries (Figure 3a), and its specific capacity was found to be approximately 40% higher than that of conventional graphene/sulfur composite cathodes (Figure 3b). This indicates that rational electrocatalysis can improve the performance of Li-S batteries. The modification of Au NPs on commercial acetylene black/sulfur composite cathodes (CB-S-Au) is also capable of electrocatalysis [74]. Rapid conversion of polysulfides is achieved by regulating the chemistry between Au NPs and polysulfides, thus improving the performance of Li-S batteries (Figure 3c,d). However, noble metal materials are expensive and difficult to use for large-scale production of Li-S batteries [40,75]. Therefore, exploring cost-effective electrocatalysts for Li-S batteries is among the current research hotspots. Metal compounds, which are abundant in yield and relatively low cost, have been widely used in the field of Li-S battery electrocatalysis as substitutes for noble metals.

#### 2.1.1. Metal Oxide

As the most common metal compounds, metal oxides are widely used in the field of Li-S battery electrocatalysis [76,77,78,79,80]. Metal oxides have large electrochemically active surfaces and contain hydrophilic groups. At the same time, the oxygen atom is a good electron acceptor, and it is easy to chemically bond with lithium [81,82]. This can increase the binding energy with polysulfides and decrease the kinetic barrier for Li_2_S nucleation, which is beneficial to the deposition and transformation of Li_2_S [83]. In addition, the cost of general metal oxides is relatively low, which is convenient for the large-scale application of catalysts.

Iron-based oxides are among the most common types of metal oxides. Loading α-Fe_2_O_3_ NPs on three-dimensional (3D) porous graphene (Figure 4a) as the cathode of Li-S batteries can achieve electrocatalytic effects [77]. The 3D structure can increase the contact area of the electrocatalyst and improve the reaction kinetics (Figure 4b). In addition, there is a strong interaction between Fe_2_O_3_ NPs and polysulfides, which can accelerate the rapid conversion of polysulfides to insoluble Li_2_S and effectively suppress the shuttle effect. In addition to 3D structures, one-dimensional (1D) nanostructures can also promote electrocatalysis. The introduction of Fe_3_O_4_-encapsulated NPs into 1D carbon nanotubes (CNTs) as hosts for Li-S batteries can effectively accelerate the electrochemical reactions inside the batteries [83]. Due to the large inner cavity of the Fe_3_O_4_-encapsulated NPs, it can ensure the cathode material with high sulfur content and alleviate the shuttle of polysulfides. At the same time, the outer 1D CNTs can promote fast ion/electron transport, thereby improving the redox kinetics (Figure 4c). In addition to being used in the cathode, Fe_3_O_4_ can also be used as a separator modification material for Li-S batteries to inhibit the shuttle of polysulfides and accelerate the redox kinetics [37]. Zero-dimensional (0D) ultrafine Fe_3_O_4_ nanocrystals were introduced in situ into carbon nanospheres as electrocatalysts to modify the separator, which can effectively enhance the electrochemical performance of Li-S batteries [84]. The 0D ultrafine Fe_3_O_4_ nanocrystals can be uniformly dispersed in carbon nanospheres, and can adsorb polysulfides and promote the further transformation of polysulfides (Figure 4e–g). The Li-S battery with the modified separator has a capacity of 1210 mAh g^−1^ at 0.2 C and a cycle decay rate of 0.027% for 1000 cycles at 1 C (Figure 4h,i).

In addition to iron-based oxides, dioxides such as MnO_2_ and TiO_2_ have also been used in the electrocatalysis of Li-S batteries [85,86,87,88]. Two-dimensional (2D) ultrathin MnO_2_ nanosheets were shown to accelerate the conversion of polysulfides to insoluble Li_2_S/Li_2_S_2_ [86] (Figure 5a). Different from the adsorption–catalysis principle, this process is mainly due to the in situ reaction of polysulfides on the 2D ultrathin MnO_2_ surface to generate thiosulfate. The surface thiosulfate groups form active polythionates complexes in subsequent reactions and acts as anchoring and transfer mediators to inhibit the dissolution of polysulfides into electrolytes (Figure 5b,d). The planar structure of 2D materials is more favorable for the deposition of reactants. Li-S batteries with sulfur/MnO_2_ nanosheet composites as cathodes exhibit high specific capacity and good cycling stability (Figure 5c). At the same time, it is also demonstrated that the mechanism is applicable to graphene oxide, which has broad applicability in many 2D materials. Titanium-deficient anatase TiO_2_ (TDAT) as a separator modification material can also improve the electrochemical performance of Li-S batteries. [89]. The TDAT-modified separator can not only adsorb polysulfides, but also reduce the reaction energy barrier and catalyze the redox reaction inside the battery (Figure 5e,f). It is worth mentioning that TiO_2_ has abundant tunable nanostructures, which facilitates the construction of various heterostructures. We will describe this structure in detail in Section 3. In addition to TiO_2_, nano-Magnéli-phase titanium oxide (Ti_n_O_2n−1_) also occupies an important position in the electrocatalysis of Li-S batteries. During the electrochemical reaction, Ti_n_O_2n−1_ is capable of bonding or electrocatalysis with sulfur, which has a significant improvement on the battery performance [90,91]. In addition to common oxides, La_2_O_3_ [92], WO_3_ [93], CeO_2_ [94], V_2_O_5_ [95], and ZrO_2_ [96] are also used in the field of electrocatalysis for Li-S batteries. Abundant metal oxide species and diverse nanostructures greatly broaden the choice of electrocatalysts and promote the further development of Li-S batteries.

#### 2.1.2. Metal Nitride

Metal nitrides have also attracted extensive attention in the field of Li-S battery electrocatalysis due to their unique electronic structures, excellent chemical stability, and outstanding mechanical strength [97]. Unlike metal oxides, metal nitrides tend to have better electrical conductivity, which facilitates the transport of ions/electrons [98]. Meanwhile, abundant nanostructures such as face-centered cubic, hexagonal closed packed, and simple hexagonal also endow metal nitrides with unique physical/chemical properties [99] (Figure 6a–c). Transition metal nitrides, including VN [100], TiN [101], and WN [102], have been widely used in the field of electrocatalysis for Li-S batteries.

VN is a material with high catalytic activity, and its catalytic activity is comparable to that of noble metal materials. The composite of VN nanoribbons with porous graphene and used as cathode materials for Li-S batteries can effectively accelerated the reaction kinetics inside the batteries [100] (Figure 7a). VN possesses both high electrical conductivity and catalytic properties, enabling the rapid reaction of polysulfides on its surface (Figure 7b,c). The compatibility of 2D graphene with 2D VN nanoribbons also greatly improves the mechanical strength of the material and enables self-supporting of the cathode. In addition to VN, vanadium nitride oxides (VO_x_N_y_) have also received a lot of attention recently [103]. Unlike other electrocatalytic materials, VO_x_N_y_ exhibit a redox potential window between their oxide counterparts, around which polysulfides are able to form polysulfide complexes and enhance kinetics and polysulfide anchoring through the V–N and V–O interfaces. TiN is a metal nitride with high mechanical properties and is often used as a coating material. In recent years, with the deepening of research, the excellent catalytic activity of TiN has been found, and thus it has also been used in the field of electrocatalysis [104]. Compared with its corresponding oxide TiO_2_, TiN has a higher electrical conductivity, which means that sulfur can be used more efficiently. Therefore, in the field of Li-S battery electrocatalysis, TiN often shows superior performance to TiO_2_. Mesoporous TiN with high specific surface area synthesized by solid–solid phase separation method used zinc titanate as raw material exhibited good electrochemical properties [101] (Figure 7d). The high specific surface area facilitates the absorption of the electrolyte, and the mesopores of 2–5 nm can effectively load sulfur while confining the shuttle of polysulfides (Figure 7e,f). Li-S batteries with TiN as cathode exhibit a higher discharge capacity due to their unique nanostructure (Figure 7g). Similarly, WN is also a new type of catalytic material, whose surface properties and catalytic performance are similar to those of noble metals. Therefore, the introduction of WN with different nanostructures into various types of carbon materials can exhibit good electrocatalytic properties [102] (Figure 7h,i). The WN nanoshells have strong adsorption/catalysis ability for polysulfides, while the WN nanorods have strong electron transfer ability. Relying on this unique nanostructure, the cathode material can achieve a capacity retention rate of 80% after 500 cycles at 0.5 C with a sulfur loading of 3 mg cm^−2^ (Figure 7j). This shows that different structures of the same material will also have different effects on performance. Therefore, the rational design of nanostructures according to the demand plays an important role in the Li-S battery electrocatalysis.

#### 2.1.3. Metal Sulfides

Metal sulfides, as a class of effective catalysts, are widely used in anticorrosion, hydro-desulfurization and solar cells [105,106]. In recent years, metal sulfides have also been widely used as electrocatalysts for Li-S batteries due to their high chemical stability of sulfur [40]. Metal sulfides can form active complexes through chelation with polysulfides, thereby reducing the solvation degree of polysulfides [107]. At the same time, metal sulfides can also promote the redox kinetics inside Li-S batteries. In addition, metal sulfides with nanostructures also have the advantages of low redox potential and high specific capacity [108]. A Co_9_S_8_ with interconnected graphene-like nanostructures was synthesized and demonstrated to interact strongly with polysulfides when used as a cathode (Figure 8a) [109]. Compared with conventional porous carbon, the cyclability of the battery is improved by 10 fold when Co_9_S_8_ with a hierarchical nanostructure is used as the cathode (Figure 8b). More importantly, after the adsorption of polysulfide by Co_9_S_8_, cobalt and sulfur can act as “thiophilic” and “lithiophilic” sites, respectively, to undergo an internally couped charge transfer process, thus achieving higher electrical conductivity than metal oxides (Figure 8c,d). This proves that some metal sulfides with high electrical conductivity can be used as cathode materials for Li-S batteries. In addition, some metal sulfides can achieve electrocatalysis by forming redox intermediates. For example, Mo_6_S_8_ [110] can be chelated with polysulfides to form Li*_x_*Mo_6_S_8_ (Figure 8e). Li*_x_*Mo_6_S_8_ has multiple redox pairs in the working voltage range of Li-S batteries, and the Gibbs free energy is much smaller than the conversion of polysulfide itself (Figure 8g). Therefore, the additional reaction pathway of Li*_x_*Mo_6_S_8_ with polysulfides can promote the uniform growth of Li_2_S with little overpotential (Figure 8f). Notably, MoS_2_ with sulfur deficiencies (MoS_2−x_) has received much attention due to its high electrochemical activity associated with the presence of sulfur deficiency [111]. Compared with MoS_2_, MoS_2−x_ can participate in the reaction with polysulfides more effectively and greatly enhance the conversion kinetics of polysulfides [112].

With the increasing abundance of synthetic methods for 2D metal sulfides in recent years, more and more metal sulfides with special nanostructures have been used for electrocatalysis in Li-S batteries. Metal dichalcogenides, as a class of metal sulfides with 2D layered structures, have been used in Li-S battery electrocatalysis due to their highly exposed surface atoms and high electrical conductivity of metalloids. Due to the unique 2D structure, the structural edge and surface of metal dichalcogenides often exhibit different electrocatalytic activities. Therefore, there is a difference in the properties [113] between the unsaturated-coordinated edge metal sites and the saturated-coordinated surface metal sites in MoS_2_ (Figure 9a,b). The adsorption capacity of the edge unsaturated sites was found to be much larger than that of the surface saturated sites, which proved that the adsorption capacity was affected by the chemical environment. In addition, CoS_2_ [114] was also added as an electrocatalyst to the carbon/sulfur cathode to improve the performance of Li-S batteries by accelerating the redox kinetics of polysulfides. CoS_2_ can form chemical bonds with polysulfides and donate electrons to them (Figure 9c,d). This accelerates the electron transfer rate from the current collector to the polysulfide, so that the reduction process is no longer controlled by the charge transfer rate. FeS_2_ is also used as an electrocatalyst in Li-S batteries. Monodisperse sub-10 nm FeS_2_ [115] nanoclusters were modified on porous carbon surfaces for enhanced electrochemical kinetics. FeS_2_ accelerates the diffusion of ions and improves the solid–solid transformation kinetics (Figure 9e,f). To further explore the potential of metal sulfides for electrocatalysis in Li-S batteries [116], the ability of various metal sulfides to adsorb polysulfides and oxidize Li_2_S was systematically explored (Figure 9g). Experiments demonstrated that VS_2_, TiS_2_ and CoS_2_ exhibit better electrocatalytic activity, which was consistent with the density functional theory simulation results (Figure 9h). This illustrates that theoretical calculations can serve as a guiding principle for screening advanced electrocatalytic materials.

#### 2.1.4. Metal Phosphides

In the periodic table, almost all transition metals can react with phosphorus to form metal phosphides [117]. Flexible components and structures make metal phosphides often exhibit excellent properties such as magnetism and electrical conductivity. In addition, many metal phosphides have excellent properties similar to those of metal dichalcogenides due to similar bonding methods [118]. This enables the application of metal phosphides in the electrocatalysis of Li–S batteries. However, it is worth noting that the catalytic site of metal phosphide is not in its bulk phase, but at the oxidized compound on its surface [58]. To achieve efficient exposure of active sites to increase electrocatalytic performance, various nanostructures have been developed and utilized. In 2018, Co-O-P was found to be formed in surface-oxidized CoP [119], which enables Co sites to interact strongly with polysulfides and generate Co-S bonds (Figure 10a). For the unoxidized CoP, the Co and P sites cannot interact with polysulfides due to the weak polarization of the Co-P bond. Therefore, the oxidized CoP has better redox kinetics than the unoxidized CoP (Figure 10b). In addition, the special structure of the carbon nanoarray-coated FeP materials [120] can facilitates the fast conduction of ions/electrons (Figure 10c). Meanwhile, the presence of Fe-O and O-P bonds in the XPS spectra indicated that the FeP nanoarrays were oxidized (Figure 10d). The oxidized FeP has a strong interaction with polysulfides, which accelerates the reaction kinetics. Due to the complex electrocatalytic properties, it is necessary to explore in depth the electrocatalytic effect of various types of transition metal phosphides on Li-S batteries [121]. Ni_2_P, Fe_2_P and Co_2_P are all able to adsorb polysulfides and accelerate the conversion of polysulfides to different degrees. Although the adsorption energy of Fe_2_P is the highest, too high adsorption energy may lead to the deposition of polysulfides on the cathode surface, which is not conducive to the continuation of the electrochemical reaction (Figure 10e,f). Therefore, a reasonable and moderate adsorption energy can maximize the catalytic effect of metal phosphides.

Metal compounds, as substitutes for noble metals, have been widely used in the field of electrocatalysis of Li-S batteries. Various types of metal compounds with excellent properties are continuously developed and utilized. However, all kinds of metal compounds have their own limitations, such as the low conductivity of metal oxides and the difficulty of preparing metal sulfides. Therefore, rational design of nanostructures is a feasible way to improve the electrocatalytic performance of metal compounds [122,123].

### 2.2. Heterostructures

Although a single type of electrocatalyst shows a certain improvement in electrochemical performance, its properties such as electrical conductivity, specific surface area and nanostructure are relatively fixed, which makes the modification of materials difficult to achieve and adversely affects battery performance. In addition, it is difficult for single-component electrocatalysts to combine the two functions of efficient adsorption and rapid conversion. Therefore, the design of heterostructure electrocatalysts composed of two or more components can overcome the limitation of the function of a single component [124,125,126]. The synergistic effect of multi-component materials to exert different functions can further improve the performance of Li-S batteries [41]. It is worth noting that the multi-component materials here include not only the metal compounds mentioned above, but also conductive polymers and other materials [44,127].

#### 2.2.1. Metal-Based Heterostructures

Metal compounds have received a lot of attention in the electrocatalysis of Li-S batteries due to their diverse types and wide sources. However, single metal compounds often have their limitations, and various properties cannot be optimal at the same time. In response to this problem, the concept of bifunctional heterogeneous structure was proposed [128]. The combination of metal oxides with high adsorption and metal nitrides with high conductivity enables the effective preparation of heterogeneous structured electrocatalysts (Figure 11a,b). The combination of TiO_2_ for the adsorption of polysulfides and TiN for catalytic polysulfide conversion enables an efficient adsorption–catalysis process. The heterostructure not only improves the poor conductivity of metal oxides, but also improves the adsorption of metal nitrides. Based on this idea, a checkerboard-like heterostructure electrocatalyst [129] was designed by the sulfonation reaction (Figure 11c). The electrocatalyst is composed of CoS_2_ and Co, and the CoS_2_-Co heterostructure has strong adsorption and catalytic effects on polysulfides (Figure 11d). In addition, the checkerboard-like nanostructure formed by porous carbon and CoS_2_-Co structure can provide abundant pore structure, which can accommodate sulfur and form a 3D conductive network (Figure 11e). As a 2D layered metal compound with high electrical conductivity, MXene has been widely used in the field of energy storage in recent years [130,131,132]. A Ti_3_C_2_T_x_/TiO_2_ heterostructure catalysts was successfully prepared by in situ oxidation [133] and used as interlayers (Figure 11f). The interlayers can effectively inhibit the shuttle effect and improve the cycling stability of the Li-S batteries (Figure 11g). The Li-S batteries with a heterostructure interlayers exhibited an average capacity decay rate as low as 0.028% after 1000 cycles at 2.0 C, which is significantly better than that of single-component interlayers (Figure 11g).

#### 2.2.2. Carbon-Based Heterostructures

Carbon materials have become among the most used substrates in heterostructures because of their diverse types, rich structures, good conductivity, and large specific surface area [41,134]. Common carbon-based heterostructures refer to hybrid materials that combine carbon materials with other materials (mainly metal compounds). Commonly used carbon materials mainly include CNTs, graphene and activated carbon [135,136]. Compared with metal-based heterostructures, carbon-based heterostructures have more abundant synthesis methods and more diverse structures. In addition, the good processability of carbon materials also provides a broader space for the application of carbon-based heterostructures. Aiming at the problem of good adsorption but poor conductivity of VO_2_, in situ growth of VO_2_ on the surface of 2D reduced graphene oxide (rGO) by solvothermal method (Figure 12a) to construct heterostructures is a good solution [137]. Compared with the single component, the reaction kinetics of VO_2_@rGO is significantly improved, and the diffusion of Li^+^ is smoother (Figure 12b). This is achieved by VO_2_ with catalytic properties and 2D rGO with good conductivity, which reflects the superiority of the heterostructure. A heterostructure composed of CNTs and MoP_2_ NPs was combined and applied to the interlayer of Li-S batteries [138]. Benefit from the synergistic effect of the heterostructure, the cycle performance of batteries is improved (Figure 12c,d). In addition to binary heterostructures, carbon-based ternary heterostructures also show good properties. The 3D network structure composed of CNTs and rGO exhibits considerable advantages in the fast transport of Li^+^ and the loading of S. Based on this idea, a 3D porous WS_2_-rGO-CNTs ternary heterostructure was designed [139], in which rGO is chemically bonded to CNTs and WS_2_ is grown in situ on CNTs (Figure 12e,f). The 3D structure shows a good mechanical property and a fast Li^+^ transport network, and the WS_2_ loaded inside serves as active sites to accelerate the catalytic conversion of polysulfides (Figure 12g). In addition to the above-mentioned carbon materials with different nanostructures, carbon materials containing heteroatoms are also able to adsorb polysulfides and accelerate their conversion due to their polar functional groups [140,141]. Carbon materials containing N, O, P, S tend to exhibit stronger adsorption–catalysis effects than pure carbon materials. Taking nitrogen-doped carbon materials as an example, carbon nitride [142] or carbon materials containing nitrogen [143] can effectively anchor polysulfide to nitrogen-containing active sites, thus greatly improving electrochemical properties.

#### 2.2.3. MOFs/COFs-Based Heterostructures

Metal-organic frameworks (MOFs) are coordination polymers self-assembled by transition metal ions and organic ligands containing oxygen, nitrogen, and other elements [144]. As a new class of porous materials, MOFs have attracted much attention in the field of energy storage due to their tunable nanostructure and functionalization [145,146]. MOFs-based heterostructures are MOFs carbonized in an inert environment to generate MOFs-derived carbon-metal framework structures with high porosity. The framework structure can be customized according to the nanostructure of the precursor, and simultaneously contains heteroatoms and metal ions. This enables MOFs-based heterostructures to exert dual confinement and catalytic effects on polysulfides. Carbonization of MOF precursors is a common method to prepare MOF-based heterostructures. For example, 3D porous cobalt-N-doped graphitic carbon (Co-N-GC) heterostructures can be obtained by carbonizing Co-MOF (Figure 13a) [147]. The uniform dispersion of Co NPs in the heterostructure improves its catalytic effect, while the 3D porous N-GC substrate can effectively adsorb polysulfides (Figure 13b). Benefiting from the advancement of the MOF-based heterostructure, the high sulfur cathode (70%) can operate stably for more than 500 cycles at 1 C (Figure 13c). Based on the interaction between MOF and MXene [148], bimetallic selenides on nitrogen-doped MXene (CoZn-Se@N-MX) enable successful self-assembly (Figure 13d). The combination of 0D CoZn-Se NPs and 2D N-MX nanosheets provides amphiphilic lithium-sulfide binding sites that accelerate redox kinetics. In addition, the 0D-2D heterostructure also has a hierarchical porous structure with a large specific surface area, which can effectively adsorb and catalyze the conversion of polysulfides and improve the cycle stability of the battery (Figure 13e–g).

Covalent organic frameworks (COFs) are novel porous crystalline materials with multi-dentate organic structures connected by covalent bonds, which have the advantages of large specific surface area, controllable functional groups, low framework density, open pore structure and diverse nanostructures [149]. Based on the above advantages, COFs have important application prospects in the field of energy storage [150,151]. Compared with traditional inorganic porous materials, the main advantage of COFs is that they have good nanostructure tunability. More and more COFs have been designed and applied with the structure from 0D to 3D and the pore size from micropore to mesopore. Heterostructure materials based on COFs have also received extensive attention. Homogeneous embedding of anatase/rutile TiO_2_ nanodots (10 nm) into porous COFs enables to obtain a hybrid-crystal-phase TiO_2_/COFs heterostructure (Figure 14a,b) [152]. This unique heterostructure can anchor polysulfides and improve reaction kinetics when used as a sulfur host material. In addition, calculations were able to demonstrate that the binding energy of different phases of TiO_2_ on polysulfides were different. Specifically, long-chain polysulfides are more prone to combine with rutile-TiO_2_ (110), while short-chain polysulfides are more prone to combine with anatase (Figure 14c). This shows that the different nanostructures have a certain impact on the electrocatalytic performance. The composite of COFs and MOFs porous materials can also form heterostructures. Wrapping the porous MOFs with microporous COFs enables to obtain heterogeneous structures with advanced nanostructures [153], where the inner MOFs act as catalysts to accelerate the conversion of polysulfides, while the outer COFs layer restricts the shuttle of polysulfides through polar functional groups (Figure 14d). The synergistic effect of COFs and MOFs can significantly improve the rate performance and cycle stability of the battery.

Heterostructures have attracted much attention due to their tunable nanostructures and excellent electrocatalytic properties. However, the complex synthesis process and the complicated design ideas make it difficult for the heterostructure to be applied on a large scale. In addition, the catalytic active sites of both metal compounds and heterostructures are limited. Limited catalytic sites may be at risk of failure during long-term cycling of Li-S batteries. Therefore, it is necessary to explore advanced nanostructures to endow electrocatalysts with more active sites.

### 2.3. Single Atoms

As mentioned previously, defect-rich metal compound electrocatalysts as well as heterostructure electrocatalysts with complementary properties can provide a large number of active sites for enhancing the redox kinetics of Li-S batteries, thus significantly improving the overall electrochemical performance [154,155]. However, the relative content of defects limits the specific catalysis activity of these catalysts, and affects the further improvement of rate performance and energy density. Therefore, making the catalyst more defective while reducing its overall size can form more active sites and increase the specific activity of the catalyst [156]. Based on this theory, single-atom catalysts (SACs) obtained by combining individual metal atoms with catalyst supports can achieve the most efficient utilization of catalyst specific surface area and metal atoms, and their adverse effects on energy density can be minimized due to the light weight of SACs [157,158,159], which together prove that the SACs is a promising research direction in the field of Li-S battery electrocatalysis.

Notably, the uniformly distributed isolated atoms, although having extremely small particle size, have a tendency to aggregate into clusters or particles due to their thermodynamic instability, which can reduce the specific catalytic activity of the catalyst. For this reason, the interaction between isolated atoms and supports was proposed in 1978, through which the movement and aggregation of isolated atoms can be prevented and a uniform and stable distribution of active sites can be constructed, thus positively affecting the selectivity, stability and electrocatalytic activity of the catalysts [160,161,162,163]. Therefore, the selection of coordination configurations with appropriate metal-atom–support interactions is also crucial for the practical application of SACs.

This section introduces the composition and coordination configuration of SACs, and summarizes some corresponding research results, aiming to provide some references on the principles and ideas for the research in this field.

#### 2.3.1. Saturated Coordination Configuration

Typical SACs usually use carbon materials as supports in which they are coordinated with other atoms and form saturated coordination configurations with symmetric electron distribution (most commonly the M-N_4_ configuration, M stands for metal atoms and N for coordination atoms such as N, O and S atoms) [164,165]. SACs with saturated coordination configurations were first applied to Li-S batteries in 2018, when single-atom Fe (SAFe) coordinated with four pyrrole N on the surface of the supports was used as an electrocatalyst to improve the rate performance and cycling lifetime of the battery, demonstrating the electrocatalytic effect of SAFe on the redox reaction of LiPSs [166]. To investigate the intrinsic mechanism of SACs in the electrochemical reaction of Li-S batteries, a novel theoretical simulation study was proposed, which demonstrated that SAFe can coordinate with Li_2_S, thus weakening the Li-S bond, promoting the decomposition of Li_2_S and the growth of LiPSs chains, and that it can regenerate by gaining electrons during the charging phase (Figure 15a) [167]. A reduction in the reaction energy barrier by SACs can effectively improve the rate performance, cycling performance and sulfur utilization of Li-S battery, thus triggering researchers to further investigate their preparation, catalytic mechanism and structural design. Generally, the catalytic activity of SACs depends on the intrinsic activity of the metal atom, the type of coordination atom, and the coordination configuration (corresponding contents are developed in Section 2.3.2), and the precise design of SACs can be achieved by modulating these factors [168,169].

As the center of the active sites provided by SACs, the catalytic activity of the metal atoms themselves and the coordination stability have a direct impact on the electrocatalytic activity of SACs. Therefore, based on the purpose of screening the most suitable metal atoms for the preparation of SACs for catalytic Li-S electrocatalysis, a novel theoretical model verified the effect of SACs constructed with different monometallic atoms in saturated coordination configurations on the electrocatalytic capacity of Li-S batteries [62]. SACs (M-N_4_ configuration, M represents Fe, Mn, Ru, Zn, Co, Cu, V, Ag) obtained by loading different monometallic atoms on a conductive support were subjected to calculations of the Li_2_S decomposition potential (Figure 15b), Li-S bond parameters (Figure 15c) and adsorption energy for Li_2_S_6_. The results show that, except for the structurally unstable SACu@NG, SAAg@NG, the VN_4_/NG-SAC group exhibits the lowest Li_2_S decomposition barrier (1.10 eV) and the highest adsorption energy for Li_2_S_6_ (3.38 eV), i.e., the VN_4_/NG-SAC has theoretically stronger electrocatalytic capacity and inhibition of the shuttle effect, which is fully consistent with the excellent discharge capacity of 1230 mAh g^−1^ (0.2 C) and the good cycling stability (0.073% capacity decay per cycle after 400 cycles) exhibited by the S@VN4/NG-SAC cathode in electrochemical tests. In addition, to deeply understand the influence of SACs on the reaction kinetics in the discharge process of Li-S battery, the Gibbs free energy calculation was performed between S-SACo@NG and S-SAV@NG with high catalytic activity in the simulation and blank control group (Figure 15d). The results showed that for the reaction rate-determining step (Li_2_S_2_→Li_2_S) with maximum positive Gibbs free energy, both SACo@NG and S-SAV@NG, which have strong ability to weaken the Li-S bond, exhibited relatively lower Gibbs free energy. Therefore, for such saturated coordination configurations of SACs, the choice of Co, V helps to achieve higher electrocatalytic activity.

As mentioned above, in order to prevent the movement and aggregation of isolated atoms, metal atoms are stably present in the support environment in various coordination configurations, so changes in the coordination environment (ligand atom type, coordination configuration, microstructure) will also determine the catalytic activity of SACs [170]. Due to the excellent structural stability and high electrocatalytic activity, N is usually chosen as the coordination atom to construct an electronically symmetric M-N_4_ saturated coordination configuration with the metal atom to achieve enhanced kinetics of the polysulfide reaction and suppression of the shuttle effect (Figure 16a,b) [171]. To deeply dissect the microstructure, the homogeneous dispersion state of Co on the support was analyzed by aberration-corrected scanning transmission electron microscopy (STEM) high-angle annular dark field (HAADF) (Figure 16c) imaging, x-ray absorption near edge structure (XANES) (Figure 16d) and extended x-ray absorption fine structure (EXAFS) (Figure 16e), which elucidate the Co-N_4_ saturated coordination conformation of such SACs at the atomic level (Figure 16f) [172]. For the coordination atoms, in addition to nitrogen atoms, heterogeneous atoms such as S, O and P are also capable of modulating the electronic structure of the central metal atom, so it is equally important to clarify the effect of choosing coordination atoms on the catalytic activity for the construction of high performance SACs.

The SACs constructed by co-coordination of N, S atoms also exhibited significant superiorities in accelerating the Li-S electrochemical reactions, for example, single-atom catalysts with Co-N_3_S-C coordination structures prepared by thiourea (Figure 17a) [173]. Compared with Co-N_4_-C, the Co-N_3_S-C configuration containing S atoms has stronger “lithiophilic” sites, which can effectively enhance the adsorption capacity of polysulfides (Figure 17b,c). More importantly, the introduction of S as a coordination atom changed the original Co-N_4_-C coordination configuration of Co, provided sufficient anchor sites for CoSAs, and significantly increased the number of active sites provided by CoSA, thus effectively enhancing the catalytic activity of SACs towards (sulfur reduction reactions) SRR, which is in full consistent with the results of Tafel curves based on LSV curve fitting (Figure 17d). With the purpose of simultaneously solving the problems of lithium dendrites and shuttle effects inherent to Li-S batteries, single-atom array mimics with Co-O_4_ coordination structures constructed from periodically arranged Co and O atoms on the surface of ultrathin metal-organic frameworks were successfully applied to Li-S batteries (Figure 17e) [174]. On the sulfur cathode side, the massive array of Co-O_4_ groups distributed on the MOF can effectively trap polysulfides through Lewis acid–base interactions to suppress the shuttle effect (Figure 17f), and on the Li anode side, the strong adsorption of O atoms on the single-atom array mimic surface to Li ions can promote uniform Li conduction, and the 2D MOF diaphragm with high Young’s modulus is able to resist the puncture of lithium dendrites, which is reflected in the electrical performance in terms of long cycling lifetime (only 0.07% capacity per cycle after 600 cycles). Furthermore, as an electrocatalyst, the single-atom array mimic can effectively enhance the conversion kinetics of polysulfides and thus the utilization of active materials, as demonstrated by comparing the high-discharge plateau (Q_H_) and low-discharge plateau (Q_L_) of the batteries with/without B/2D MOF-Co separator (B/2D MOF-Co cells with capacities of 350 and 758.3 mAh g^−1^, respectively) (Figure 17g).

Based on the work of SACs, a dual-atom catalyst Co-P cluster/NC using P atoms as ligand atoms as well as catalytic progenitors (Figure 18a) was proposed, and the local structure and chemical environment of Co atoms were further analyzed by XANES and WT-EXAFS (Figure 18b), which demonstrated the Co-P cluster/NC configuration constructed by coordination of atomic Co with P, N [175]. This work demonstrates that P atoms with proper electronegativity as coordination atoms can synergize with Co atoms to effectively promote polysulfide conversion and Li_2_S nucleation (Figure 18c,d) and enhance the reaction kinetics. It is noteworthy that the effect of micromorphology on the coordination environment also directly affects the activity of SACs. The introduction of highly folded structures into the support can modify the local coordination structure of the FeN_4_ group (Figure 18e), thus achieving a simultaneous enhancement of the electrocatalytic activity of SACs at different current densities [176]. By further DFT calculations and constructing a theoretical model (Figure 18f), it is demonstrated that the number of electrons obtained in the density of states (DOS) of the d orbitals of Fe atoms decreases (Figure 18g), which proves that the microstructure of the support can directly affect the surface electronic states of the metal atoms in the coordination environment. Based on the comparison of the overpotential in the last step of the SRR reactions, it is demonstrated that the active site provided by FeNC/wG has a stronger binding capacity for LiS* (Figure 18h). Thus, FeNC/wG obtained by microstructural modulation is able to kinetically promote the slow Li_2_S redox reaction due to the atomic-level modified FeN_4_ active sites with different geometric symmetries and electronic structures.

#### 2.3.2. Unsaturated Coordination Configuration

Although saturated coordination configuration SACs, represented by M-N_4_-type structures, have been widely used in Li-S electrochemical reactions due to their structural stability and catalytic activity, unsaturated coordination configuration SACs with asymmetric electron distribution usually have greater advantages in terms of electrocatalytic activity and adsorption capacity to LiPSs. To explore the effect of the coordination number of atoms on the catalytic activity of SACs, the investigators demonstrated the stronger adsorption capacity and facilitated conversion kinetics of LiPSs for the oversaturated coordination configuration of Ni-N_5_ by a combination of DFT calculations, electrochemical tests, and spectroscopic characterization [177]. Comparison of the electron distribution of Ni atoms in the Ni-N_5_/C, Ni-N_4_/C and Ni-N_3_/C configurations shows that the Ni-N_5_/C configuration has significant d-orbital electron depletion and oversaturated N atoms can induce electron transfer to Ni atoms (Figure 19a), thus enhancing the catalytic activity for the conversion reactions of LiPSs, which fully illustrates the results of DFT fitting of energy profiles and binding energies (Figure 19b,c). Fe-N_5_-type SACs in supersaturated coordination configuration demonstrated that Fe monoatoms with high N coordination number can strengthen the immobilization of LiPSs compared to saturated coordination (Figure 19d) and effectively reduce the energy barriers for the conversion of LiPSs and Li_2_S nucleation (Figure 19e,f) [178]. In order to address the problem of insufficient adsorption capacity of M-N_4_ configuration SACs for polysulfides, Fe-N_3_C_2_-C SACs constructed by using C and N as co-coordinating atoms were designed and prepared with asymmetric supersaturated coordination configurations, which not only form additional π-bonds with the p orbital hybridization of S to anchor LiPSs, but also effectively enhance the redox kinetics of LiPSs [179]. The authors demonstrated the orbital hybridization mode by calculating the predicted density of states (PDOS) of the central Fe-atoms in Fe-N_3_C_2_-C and Fe-N_4_-C after adsorption of Li_2_S (Figure 19g,h), which explains the strong adsorption capacity of Fe-N_3_C_2_-C for polysulfides exhibited in the UV–vis tests (Figure 19i).

In addition to supersaturated coordination configurations, unsaturated coordination configurations also have good applications in the design of SACs for Li-S batteries. Single-atom catalysts with Mo-N_2_-C conformation (Figure 20a) with good adsorption of LiPSs and Li_2_S (Figure 20b) and improved conversion kinetics (Figure 20c) were applied to obtain Li-S batteries with high reversible capacity (743.9 mAh g^−1^ at 5 C rate) and long cycling lifetime (after 550 cycles with a capacity decay of only 0.018% per cycle) [180]. Through a combination of supporting structure design and coordination environment modulation, SACs with FeN_2_ unsaturated coordination structures were introduced onto graphene with a pore-rich structure (Figure 20d), achieving uniform Li transport and effective adsorption of LiPSs [181].

## 3. Effect of Nanostructure on Heterogeneous Electrocatalysts

Nanostructures are small structures with physical dimensions below 100 nm. At the nanoscale, the structure and properties of materials often change in many ways. Therefore, nanostructured materials have broad application prospects in many fields. According to the dimensional classification, nanostructures can be generally classified into 0D, 1D, 2D, and 3D. The 0D nanostructures mainly include quantum dots (QDs), nanoparticles (NPs) and nanoclusters; 1D nanostructures include nanowires, nanorods, nanotubes and nanofibers; 2D nanostructures mainly include nanosheets and nanoribbons; 3D nanostructures are usually composed of the above three dimensions of nanostructures. In the field of electrocatalysis of Li-S batteries, electrocatalysts with different nanostructures usually show different catalytic properties and advantages. Due to the large number of exposed catalytic active sites, the electrocatalysts with 0D nanostructures can improve the catalytic performance while minimizing the amount of catalyst. Such structures are common in the field of single-atom catalysis. Electrocatalysts with 1D and 2D nanostructures have characteristic morphologies and have considerable advantages in ion/electron transfer. In addition, 1D and 2D nanostructures as cathode materials can effectively inhibit the volume change during the charge–discharge process of Li-S batteries and ensure the stability of the catalytic process because of the good mechanical property. For 3D nanostructured electrocatalysts, they have large specific surface area and can be used as sulfur hosts. The 3D nanostructure can accommodate more sulfur while ensuring effective electrocatalysis. Additionally, the rich 3D structure can also improve the dispersion of elemental sulfur, which is conducive to the uniform deposition of Li_2_S and improves the poor conductivity of charge–discharge products. In addition to the above nanostructures, the heterostructure catalysts formed by the combination of various nanostructures of different dimensions can integrate the advantages of each component, realize the synergistic effect between the components, and improve the electrocatalytic performance. Based on recent research, a brief overview is provided in Table 1. The positive effects of different kinds of nanostructures on the electrocatalysis of Li-S batteries are introduced.

In particular, the electrocatalytic properties of typical cathode materials should be explored in more detail. Here, the sulfur content of the cathode, the electrocatalyst content of the cathode and the specific surface area are listed to facilitate a clearer comparison of the nanostructures of various types of non-homogeneous catalysts in relation to the capacity of the battery. It can be seen that single-atom electrocatalysts usually have a greater advantage in terms of specific surface area and catalyst usage compared to the other two types of electrocatalysts. Because single-atom electrocatalysts expose more active sites, their specific surface area tends to be larger, and their usage in the cathode is lower, enabling them to catalyze electrochemical reactions within Li-S batteries more efficiently. Various types of single-atom electrocatalysts have been designed and reported, and the performance of Li-S batteries has been gradually improved. Certainly, single-atom electrocatalysts also have the disadvantages of complicated synthesis methods and high cost. Therefore, how to synthesize single-atom electrocatalysts for Li-S batteries in large quantities in an efficient, fast, and inexpensive manner is a key issue at present. In addition, it can be seen from Table 2 that more electrocatalyst content may affect the specific surface area of the material, which may be related to excessive electrocatalyst agglomeration. At the same time, too much electrocatalyst will also reduce the energy density of Li-S batteries, which is not conducive to the practical application of Li-S batteries. Therefore, it is necessary to ensure the catalytic effect while reasonably controlling the electrocatalyst content.

## 4. Homogeneous Electrocatalysts

Although heterogeneous electrocatalysts that are capable of adsorbing polysulfides and reducing the activation energy of bidirectional conversion reactions have shown many advantages in enhancing the practical electrochemical performance of Li-S batteries, they still have some drawbacks due to the limitation of insufficient active sites and the microstructures of materials, which include low practical utilization efficiency and gradual failure in long cycles. Therefore, with the progressive research on soluble additives, homogeneous electrocatalysts that can dissolve in electrolyte and thus be in full contact with the active materials have attracted great attention.

Homogeneous electrocatalysts are in the same phase as polysulfides in Li-S batteries, which allows them to be fully exposed to electronic conductors, ionic conductors and polysulfides, and thus are expected to be free from reliance on active sites and avoid failure by gradual coverage with insulating products. During the electrochemical reaction of Li-S batteries, homogeneous electrocatalysts can achieve faster redox kinetics by constructing additional electronic pathways or combining with active materials to directly change the reaction pathways in the system.

Even though the homogeneous-type electrocatalysts completely dissolved in the electrolyte have impressive merits, the performance loss caused by internal shuttling cannot be ignored, therefore the semi-immobilized-type electrocatalysts with the characteristics of both heterogeneous and homogeneous electrocatalysts have been developed. As another type of homogeneous electrocatalysts, semi-immobilized electrocatalysts can ensure sufficient active sites at sulfur anode while achieving uniform and sufficient electrocatalytic effect, which couples the heterogeneous and homogeneous processes well, so it is a promising research direction in the field of Li-S battery electrocatalysis.

In this section, some homogeneous-type electrocatalysts and semi-immobilized-type electrocatalysts applied in Li-S batteries are reviewed (Figure 21).

### 4.1. Homogeneous-Type Electrocatalysts

As elaborated above, the redox reaction of active materials in Li-S system can only occur at the three-phase interface between the ionic conductor, the electronic conductor, and the active materials, while homogeneous-type electrocatalysts in the same phase as the active materials can effectively utilize the active sites and promote both nucleation and growth of Li_2_S, usually by building additional electronic pathways with faster redox kinetics or directly changing the reaction path. Nickel dimethoxyethane chloride adduct (NiDME) were selected as homogeneous catalysts for accelerating the redox conversion of sulfur as well as for inhibiting the shuttle effect of polysulfides [66]. For one thing, NiDME dissolved in the electrolyte phase can fully bind to the active materials and effectively reduce the activation energy of the multi-stage electrochemical reaction of sulfur, thus greatly accelerating the redox reaction rate (Figure 22a). For another, NiCl_2_ formed by dissociation of NiDME has higher affinity for LiPSs than for electrolytes and lower affinity for elemental sulfur than for electrolytes, thus enabling excellent recycling and effective capture of LiPSs. To verify the catalytic ability of NiDME for electrochemical processes, Tafel curve fitting (Figure 22b) and calculation of the activation energy of the discharge process (Figure 22c) were performed from the data of cyclic voltammetry tests, and these two plots demonstrated that the NiDME catalyst greatly enhanced the conversion efficiency of LiPSs through a significant reduction in the activation energy. The high catalytic activity of NiDME was further demonstrated by in situ UV tests in agreement with the theoretical calculations. It is concluded that NiDME is an active and practical electrocatalyst for trapping LiPSs, accelerating their conversion at the homogeneous interface and regulating the homogeneous deposition of Li_2_S.

Since the electrochemical reactions in Li-S batteries are reversible, in the charging process, the nucleation barrier of Li_2_S and high activation energy of the oxidation reaction of LiPSs are not negligible. Anthraquinone derivative 1,5-bis(2-(2-(2-methoxyethoxy) ethoxy) ethoxy) anthra-9,10-quinone (AQT) was incorporated into the electrolyte, which effectively enhanced the utilization of Li_2_S and prevented the deposition of “dead” Li_2_S [183]. For Li_2_S, which has very poor conductivity and is difficult to activate, AQT is able to build additional electron transportation pathways for it with its own proper redox potential during the charging process, thus promoting Li_2_S oxidation throughout the electrolyte surface and further diffusion to the current collector where it undergoes electrochemical reoxidation (Figure 22d). In order to verify the sustained redox catalytic activity of AQT over long cycles, GCD tests for AQT group as an experimental group and a control group without AQT were performed for 10 cycles (Figure 22e), and the corresponding differential capacity versus voltage (dQ/dV) curves were obtained (Figure 22f), which clearly demonstrated that the AQT group avoided voltage overcharge and only had negligible capacity decay, and that the redox peak was stable and pronounced in the AQT group over 10 cycles, respectively. Further, AQT was applied to all-solid-state Li-S batteries, using its fast electron transfer pathways constructed between active materials and current collector to greatly reduce the dissociation barrier of Li_2_S (Figure 22g), thus achieving fast Li-S redox reaction kinetics in a solid-state battery [184].

It is worth noting that homogeneous electrocatalysts possess the same phase as active materials and can make full contact with LiPSs; however, there is also an internal shuttle problem. To suppress the loss/deactivation of catalyst, the heterogeneous electrocatalyst CoSNC (CoS_1.097_ nanoparticles embedded in nitrogen-doped porous carbon sheets) was rationally coupled with the homogeneous electrocatalyst CoCp_2_ (cobaltocene), and their synergy was utilized to catalyze the electrochemical reaction of polysulfides (Figure 22h,i), in which the CoCp_2_-induced homogeneous deposition of Li_2_S formed a three-dimensional porous structure that facilitated the catalytic oxidation process on CoSNC (Figure 22j), while the nitrogen-doped porous carbon of CoSNC could confine CoCp_2_ to the sulfur cathode region to reduce its loss [185]. This mechanism of dual-mediator synergy provides a promising new strategy for the design of electrocatalytic systems and an effective design for the coupling of heterogeneous and homogeneous electrocatalysis.

Due to the slow redox kinetic limitations of intrinsic polysulfides, simply providing additional fast electron transport pathways does not mechanistically ameliorate therefore existing problems, and even a decrease in catalytic activity of such homogeneous catalysts may occur as the polysulfide concentration distribution changes. Based on this consideration, a diphenyl diselenide (DPDSe) was introduced into electrolyte, which could spontaneously react with the polysulfide to generate intermediate lithium phenyl selenium polysulfides (LiPhSePSs) with improved redox-mediated ability (Figure 23a), thus accelerating the polysulfide bidirectional redox kinetics in reaction mechanism [186]. To verify the improvement of the DPDSe-mediated reaction mechanism for the key rate-determining steps in the charging and discharging process, chronoamperometry tests (Figure 23b) and potentiostatic intermittent titration tech (PITT) tests (Figure 23c) were performed. Chronoamperometry curves at 2.1 V show that the DPDSe-mediated cell reaches its peak current approximately 9700 s faster than the blank cell, while its deposition capacity reaches 2.2 fold that of the latter, demonstrating the effective acceleration of Li_2_S nucleation kinetics by DPDSe. PITT tests, on the other hand, show that DPDSe-mediated cells exhibit earlier peak currents and higher conversion capacities for both the Li_2_S dissolution peak at 2.16 V and the S_8_ precipitation peak at 2.4 V, demonstrating the key role of DPDSe in promoting SER kinetics. The tests described above demonstrate that the DPDSe-mediated reaction mechanism of changing intermediates effectively improves the bidirectional redox kinetics of the Li-S battery, which is also in good agreement with the theoretical calculations.

Along the line of changing reaction mechanism, homogeneous electrocatalyst allyl methyl disulfide (AMDS)-mediated all-liquid-phase mechanism were further applied to Li-S batteries over a wide temperature range (room temperature to Cryogenic temperature) [187]. In the electrochemical reactions, elemental sulfur reacts preferentially with AMDS and predominantly generates the highly solvated trisulfide intermediate CH_2_=CHCH_2_SSSCH_3_, as confirmed by spectroscopic studies (in situ UV–vis and non-in situ ^1^H NMR) and molecular dynamics (MD) simulations, followed by direct reduction to the final product in the liquid phase (Figure 23d). Compared with the slow “solid–liquid–solid” reaction route, this fast all-liquid-phase reaction route significantly reduces the activation energy of the reactions and improves the redox kinetics. In addition, the introduction of AMDS was able to modulate the deposition form of Li_2_S. Current–time transients at 2.08 V (Figure 23e) and electrochemical nucleation model simulation results (Figure 23f) were obtained based on current–time transients (CTTs) tests, which visually indicate that the AMDS-mediated electrochemical reaction has higher peak nucleation current and its nucleation model conforms to the three-dimensional deposition form, demonstrating the significant acceleration of Li_2_S nucleation kinetics by the all-liquid-phase mechanism. Based on the above, AMDS-mediated cells can operate well at temperatures as low as −60 °C, demonstrating that homogeneous electrocatalysts with altered reaction mechanisms are an important guide for electrocatalyst design over a wide temperature range.

### 4.2. Semi-Immobilized-Type Electrocatalysts

In Li-S batteries, the redox reaction of intrinsic polysulfides is a multi-step reaction involving solid–liquid–solid phase changes with both heterogeneous and homogeneous processes. From this perspective, catalysts capable of regulating both processes simultaneously and coupling heterogeneous electrocatalysis with homogeneous electrocatalysis capabilities, are considered as a promising research direction for accelerating redox kinetics.

Starting from the design described above, a semi-immobilization strategy for the copolymerization of small-molecule imides with medium-chain polyethers to form space-constrained but catalytically active semi-immobilized RMs (PIPEs) was firstly proposed [188]. In PIPE-mediated electrochemical reactions, imides with appropriate redox potentials can achieve sufficient electrocatalytic acceleration of the active species by chain swinging, while polymer macromolecules do not suffer losses caused by shuttle effects due to the semi-constrained nature of polyethers (Figure 24a). By injecting PIPE into a fully discharged Li-S battery and further discharging it to 1.7 V, the result that PIPE can release an additional 140 mAh g^−1^ specific capacity was obtained, which directly demonstrates the excellent electrocatalytic ability of PIPE to enable full electrochemical reduction of unreacted polysulfides (Figure 24b). In addition, the high charging potential barrier of Li_2_S is also an inherent problem of Li-S electrocatalysis, for which Li_2_S cathode batteries with and without PIPE were tested separately for charging and discharging (Figure 24c), and the results showed that the PIPE-modified cathode could be effectively charged up to 600 mAh g^−1^ compared to the blank cell with very low capacity due to the high initial charging potential, which demonstrated the efficient electrochemical acceleration of the SER process of Li_2_S by PIPE. Thus, the PIPE that both promotes adequate polysulfide discharge and reduces barriers to Li_2_S dissociation demonstrates the compelling role of the semi-immobilization strategy for achieving bidirectional electrocatalysis.

Based on the semi-immobilization strategy, the semi-immobilized electrocatalyst were further extended to heterogeneous electrocatalysis, which verified the feasibility of coupling homogeneous electrocatalysis with non-homogeneous electrocatalysis [182]. Specifically, G@ppy-por electrocatalysts with semi-fixed active sites were prepared by covalently grafting porphyrin molecules onto graphene collectors and using polypyrrole linkers as active sites (Figure 24d). From the LSV curves, it can be seen that the G@ppy-por electrocatalyst-mediated cell exhibited the highest current response in both the kinetic and diffusion control regions during the SRR process (Figure 24e), demonstrating the potent trapping and kinetic acceleration of the polysulfide by the porphyrin active site. In addition, the G@ppy-por electrocatalyst set also exhibited the fastest response and highest peak current in the PITT test (Figure 24f), which was attributed to the high catalytic activity of the catalyst for the Li_2_S nucleation and deposition process.

In conclusion, the semi-immobilized catalysts appropriately coupled heterogeneous and homogeneous electrocatalysis, combining the performance characteristics of both, show promising prospects in accelerating the redox kinetics of Li-S batteries. Therefore, further materials research and mechanistic exploration in the field of semi-immobilized electrocatalysts will help to achieve ideal high-efficiency Li-S battery electrocatalytic reactions.

## 5. Effect of Molecular Structure on Homogeneous Electrocatalysts

Unlike heterogeneous catalysts that accelerate redox kinetics by modulation of material morphology and nanostructure, homogeneous catalysts are usually designed to enhance the kinetics of Li-S electrochemical reactions through molecular structure. Since different molecular structures lead to different catalytic mechanisms to electrocatalysts, understanding the specific roles of different molecular structures will be a profound guide to the design of Li-S battery electrocatalysts. In this section, some classical molecular structures of electrocatalysts are reviewed and their catalytic mechanisms are concisely summarized.

### 5.1. Building Fast Electronic Pathway Types

Such electrocatalysts in Li-S batteries are usually able to construct electron exchange paths with accelerated kinetics between the active material and the current collector and induce three-dimensional deposition of Li_2_S, and such catalytic mechanisms are usually determined by the following molecular structures: imide structures [188,189], anthraquinone structures [183,184], porphyrin structures [182], transition metal metallocenes, etc. [185,190].

Take PIPE, an electrocatalyst with an imide structure, as an example: in the electrochemical reaction, PIPE exhibits a two-electron reaction mechanism similar to that of pyromellitic dimide due to the intrinsic reactivity of carbonyl groups in the imide segments (Figure 25a). Combining the electrochemical reaction mechanism with its CV curves shows that the reduction peaks at 2.46, 2.33 V and 2.01, 1.9 V vs. Li/Li^+^, corresponding to PIPE from neutral to monoanion (r_1_PIPE) and from monoanion to dianion (r_2_PIPE), respectively, appear, which corresponds to the entire charge–discharge potential window of sulfur in the electrolyte (usually at 2.3–2.4 V and 2.0–2.1 V) properly (Figure 25b). Therefore, PIPEs at different redox stages are able to exchange electrons with sulfur reactive materials with different equilibrium potentials through carbonyl oxygen of the acyl imide structure, thus effectively accelerating the redox kinetics.

### 5.2. Change the Electrochemical Reaction Mechanism Types

In Li-S batteries, such electrocatalysts usually combine with active substances to form specific intermediates that directly modify the inferior redox kinetics of intrinsic polysulfides, and such catalytic mechanisms are usually determined by the following molecular structures: disulfide structures [187], phenylselenides [186], ethyl viologen structure [191], etc.

Take diphenyl diselenide DPDSe as an example: in electrochemical reactions, DPDSe reacts spontaneously with lithium polysulfide to form soluble lithium phenyl selenide polysulfide LiPhSePSs (Figure 26a) [186]. For the redox reactions of long-chain polysulfides, DPDSe exhibits excellent redox mediating ability due to the significant advantage of LiPhSePSs formed under Se–S interactions over intrinsic polysulfides in terms of the lowest unoccupied molecular orbital (LUMO) and highest occupied molecular orbital (HOMO) energy levels (Figure 26b).

In conclusion, screening suitable molecular structures to construct novel polysulfide-based electrochemical catalysts is a simple and effective strategy that can be conveniently used for practical applications. Based on recent studies, a brief overview is provided in Table 3, which presents the positive effects of different functional groups/functional structures on the electrocatalysis of Li-S batteries. Hence, a small number of suitable catalysts can effectively optimize the redox kinetics of intrinsic LiPSs and improve the utilization of active materials. Nevertheless, several key issues remain to be explored, the first of which is the elusive correlation between the optimal molecular structure of the functional catalyst and the cell performance. Secondly, the changes in the potential structure of the newly formed highly active intermediates during cycling are difficult to demonstrate. Last but not least, the introduction of excessive additives may harm the battery energy density. Therefore, in the future work, it is necessary to further develop precise molecular synthesis and structural characterization techniques to promote the research of novel high-efficiency Li-S battery electrocatalysts.

## 6. Conclusions and Prospective

With the development of the times, the design of nanomaterials with different structures for use in various fields is undoubtedly a breakthrough in science and technology. Looking back on the development path of the battery, various nanomaterials play an indispensable role in the various components of the battery. For Li-S batteries, various materials have been studied in depth to solve the problems of slow reaction kinetics and the shuttle effect. Carbon materials with special nanostructures, polymer materials capable of chemical bonding and various metal materials have been continuously developed and utilized, and the disadvantages of various materials have been continuously optimized through heteroatom doping, structural optimization, and other technologies. In addition, as a new idea, the electrocatalysis of Li-S batteries has been proposed and realized. Various metal compounds, heterostructures and single atoms have been used in heterogeneous electrocatalysis of Li-S batteries, and various homogeneous catalysts that can be dissolved in electrolytes have been developed. The successful application of these electrocatalytic materials indicates that the addition of suitable catalysts in the conventional sulfur chemistry of Li-S batteries can provide a new way for the conversion of polysulfides. However, before the large-scale application of Li-S batteries, there is still a lot of room for the design of electrocatalysts. In order to further accelerate the conversion of polysulfides and improve the electrocatalytic system of Li-S batteries, the focus of future electrocatalysts should include, but not be limited to, the following aspects:(1)Nanostructure innovation: Electrocatalysts with advanced nanostructures can effectively promote the redox of sulfur. To date, it remains a challenge to design novel structured electrocatalysts for effective inhibition of the shuttle of polysulfides. The 1D materials with fast conductivity, 2D materials with a large accessible surface, and 3D materials with a network structure can improve the electrocatalytic performance to a certain extent. Therefore, the construction of electrocatalysts with special nanostructures through advanced synthesis and preparation technologies (such as atomic vapor deposition and molecular beam epitaxy) will help to improve the performance of lithium–sulfur batteries.(2)Testing method innovation: Generally speaking, the conventional electrochemical research method is to obtain the sum of various microscopic information of the electrochemical system by means of detection with electrical signals as the excitation. In recent years, with the progress and development of nanomaterials, it is difficult for conventional electrochemical testing methods to intuitively and accurately reflect the various reaction processes, species concentrations and morphological changes at the electrode/solution interface, which brings great problems to the correct interpretation and expression of electrochemical reaction mechanism, and also limits the development of electrocatalytic materials for Li-S batteries. Therefore, the development of new testing methods, especially high-resolution electron microscopy equipment and in situ testing equipment, is helpful to understand the key role of nanostructures in electrocatalysis.(3)Exploration of catalytic mechanism: With the wide application of Li-S battery electrocatalysts, scientists have deeply realized that a series of catalysts with special nanostructures and properties can effectively regulate polysulfides. The surface reactivity, conductivity, adsorption ability of polysulfides, and catalytic conversion ability of polysulfides all affect the actual performance of Li-S batteries. However, the study of various catalytic mechanisms of electrocatalysts is far from thorough, and further exploration and relevant principles of universality are needed.

In summary, the slow and complex kinetic reaction is a major problem that hinders the practical application of lithium–sulfur batteries. The field of Li-S battery electrocatalysis provides a feasible way to solve this problem. However, looking back on the development of electrocatalysts for Li-S batteries in the past, there are still many challenges and problems to be solved in the fields of catalytic mechanism and structural innovation of future electrocatalysts. We propose a general strategy for comparing the advantages and disadvantages of heterogeneous and homogeneous electrocatalyst to discuss the above issues. We propose a general strategy for comparing the advantages and disadvantages of heterogeneous and homogeneous electrocatalyst to discuss the above issues. Since the two types of electrocatalysis have different characteristics and mechanisms, we believe that selecting an electrocatalyst based on the need for electrochemical performance is an appropriate strategy, specifically, evaluating and selecting an electrocatalyst based on the balance of electrocatalytic performance and catalytic activity retention (Table 4). We hope that this strategy can provide a reference for the subsequent research direction and selection strategy of lithium–sulfur battery electrocatalysts, and provide some help for the research of novel electrocatalysts.

## Figures and Tables

**Figure 1 nanomaterials-12-04341-f001:**
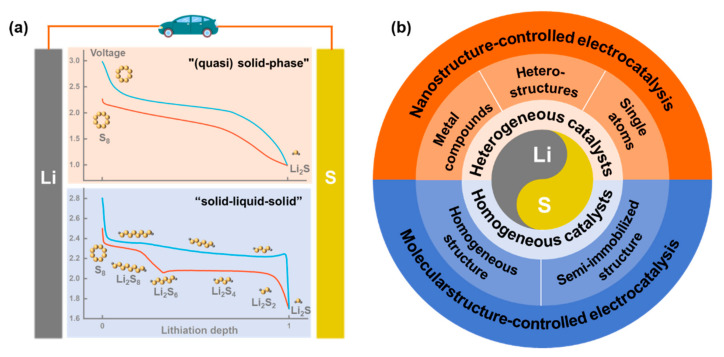
(**a**) The reaction mechanism of Li-S batteries. (**b**) Electrocatalysis in Li-S batteries.

**Figure 2 nanomaterials-12-04341-f002:**
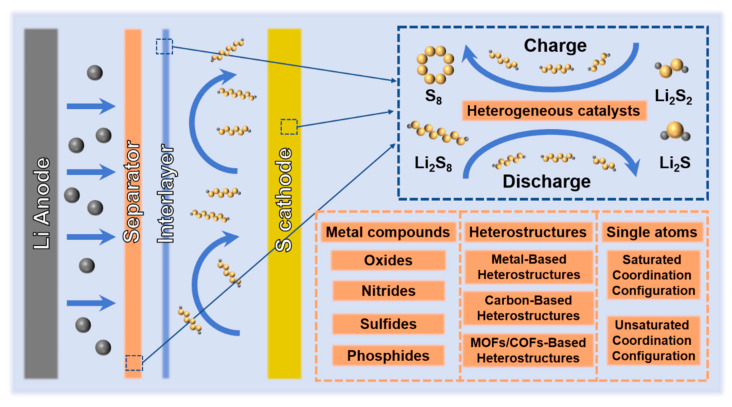
Schematic diagram of Pt-catalyzed polysulfide conversion.

**Figure 3 nanomaterials-12-04341-f003:**
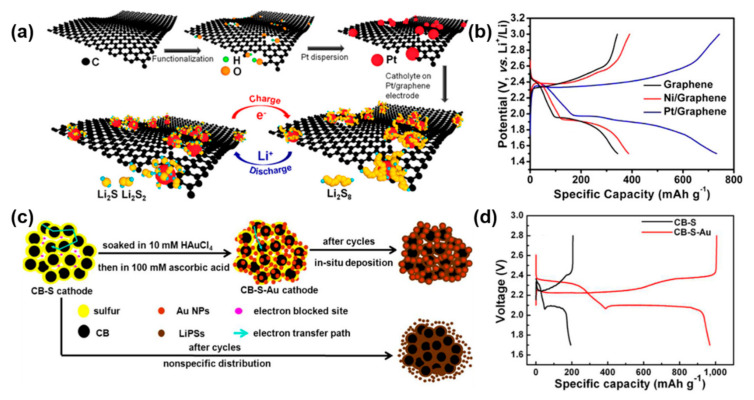
(**a**) Schematic diagram of Pt−catalyzed polysulfide conversion. (**b**) Galvanostatic charge−discharge curves of graphene, Ni/graphene, and Pt/graphene. Reprinted with permission from Ref. [57]. Copyright 2015 American Chemical Society. (**c**) Synthetic route of cathodes containing Au NPs. (**d**) Galvanostatic charge−discharge curves of CB−S and CB-S−Au. Reprinted with permission from Ref. [74]. Copyright 2015 American Chemical Society.

**Figure 4 nanomaterials-12-04341-f004:**
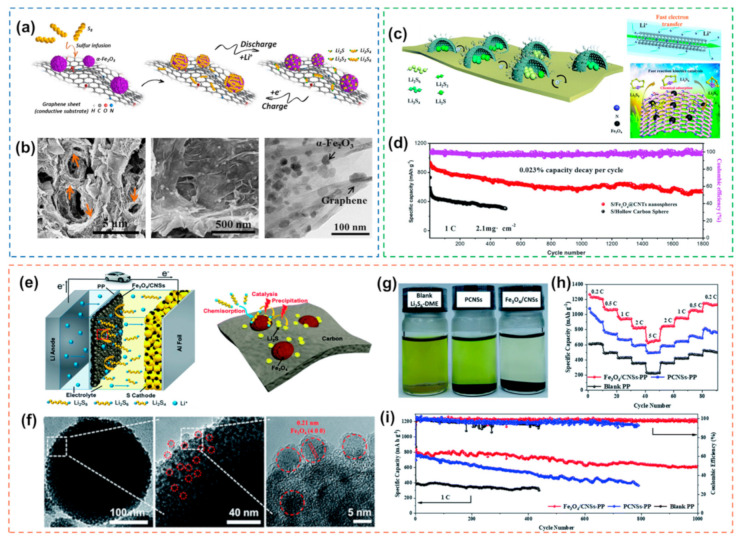
(**a**) Schematic diagram of α−Fe_2_O_3_ accelerating polysulfide conversion. (**b**) SEM images of α−Fe_2_O_3_ supported on graphene. Reprinted with permission from Ref. [77]. Copyright 2017 Elsevier. (**c**) Schematic diagram of Fe_3_O_4_−encapsulated NPs for accelerating ion/electron transfer. (**d**) Cycling performance of S/Fe_3_O_4_@CNTs nanospheres at 1 C. Reprinted with permission from Ref. [83]. Copyright 2017 Royal Society of Chemistry. (**e**) Schematic diagram of Fe_3_O_4_ modifying separator and catalyzing polysulfide conversion. (**f**) SEM image of Fe_3_O_4_ nanocrystals on the surface of carbon nanospheres. (**g**) Visualized adsorption of polysulfides by Fe_3_O_4_. (**h**) Rate capability of Fe_3_O_4_/CNSs−PP, CNSs−PP and PP. (**i**) Cycling performance of Fe_3_O_4_/CNSs−PP, CNSs−PP and PP. Reprinted with permission from Ref. [84]. Copyright 2017 Royal Society of Chemistry.

**Figure 5 nanomaterials-12-04341-f005:**
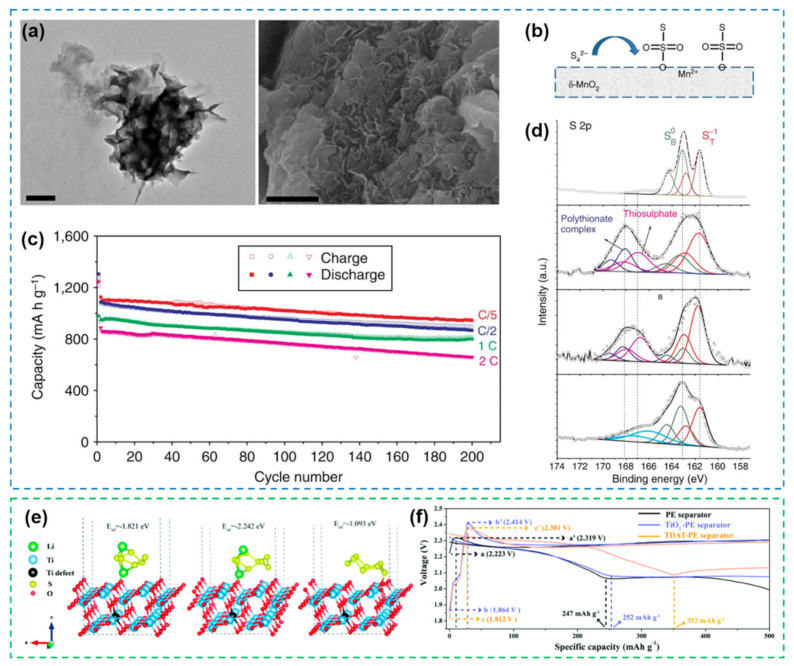
(**a**) SEM image of 2D ultrathin MnO_2_ nanosheets. (**b**) Schematic diagram of the formation of thiosulfate from MnO_2_ and polysulfides. (**c**) Cycling performance of S/MnO_2_ nanosheets at C/5, C/2, 1 C and 2 C rates. (**d**) S 2p core spectra of Li_2_S_4_, MnO_2_−Li_2_S_4_, GO−Li_2_S_4_ and graphene−Li_2_S_4_. Reprinted with permission from Ref. [86]. Copyright 2015 Springer Nature. (**e**) Density functional theory (DFT)−calculated structures and adsorption energies of Li_2_S_6_, Li_2_S_8_ and S_8_ on the TDAT (101) surface. (**f**) Charge−discharge profiles of Li−S batteries with PE, TiO_2_−PE, and TDAT−PE separators at 0.2 C with detailed information. Reprinted with permission from Ref. [89]. Copyright 2020 Royal Society of Chemistry.

**Figure 6 nanomaterials-12-04341-f006:**
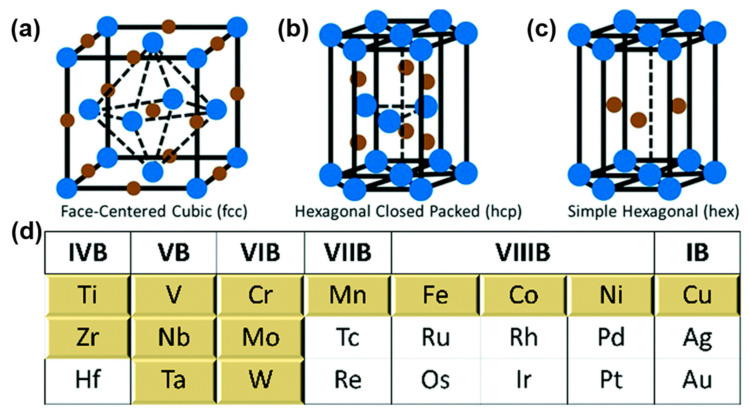
(**a**) Face−centered cubic structure model of TMNs. (**b**) Hexagonal closed packed structure model of TMNs. (**c**) Simple hexagonal structure model of TMNs. (**d**) Experimentally reported Group IVB to IB TMNs highlighted in gold. Reprinted with permission from Ref. [95]. Copyright 2016 Royal Society of Chemistry.

**Figure 7 nanomaterials-12-04341-f007:**
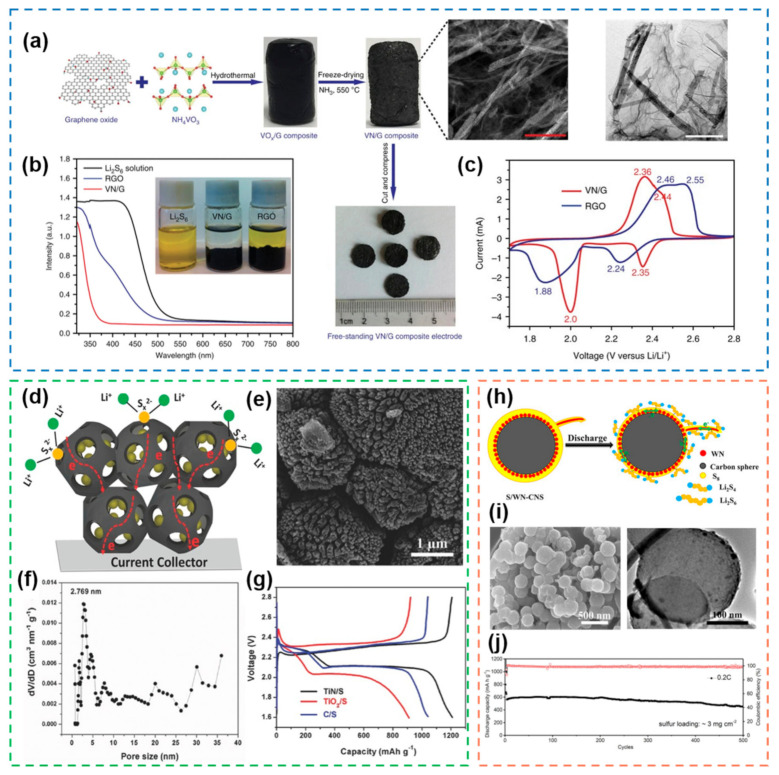
(**a**) Schematic of the fabrication of a porous VN/G composite. (**b**) UV−vis absorption spectra of Li_2_S_6_ solutions before and after addition of RGO and VN/G. (**c**) Cyclic voltammetry (CV) curve of VN/G and RGO cathodes. Reprinted from Ref. [100]. (**d**) Schematic diagram of accelerated polysulfide conversion by mesoporous TiN−S composite cathode. (**e**) SEM image of mesoporous TiN. (**f**) The pore size distribution curve of mesoporous TiN. (**g**) Galvanostatic charge−discharge curves of TiN/S, TiO_2_/S and C/S. (**h**) Schematic illustration of WN composite cathode catalyzing polysulfide conversion. Reprinted with permission from Ref. [101]. Copyright 2016 John Wiley and Sons. (**i**) SEM image of WN composite cathode. (**j**) Cycling performance of WN composite cathode. Reprinted with permission from Ref. [102]. Copyright 2019 John Wiley and Sons.

**Figure 8 nanomaterials-12-04341-f008:**
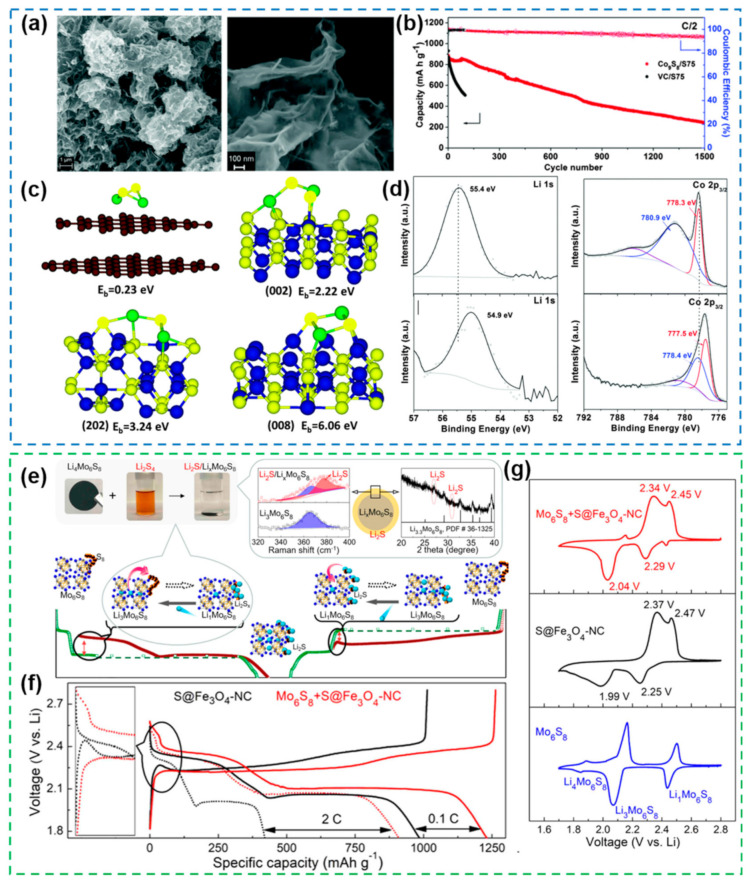
(**a**) SEM image of the as-synthesized graphene−like Co_9_S_8_. (**b**) The cycling performance of Co_9_S_8_/S75 and VC/S75. (**c**) Schematic diagram of the adsorption energy of Li_2_S_2_ on double−layer graphitic carbon and four-layer Co_9_S_8_. (**d**) High−resolution XPS spectra of the Li 1s and Co 2p_3/2_ regions of Co_9_S_8_−Li_2_S_4_. Reprinted with permission from Ref. [109]. Copyright 2016 Royal Society of Chemistry. (**e**) Redox-catalyzed sulfur conversion via Mo_6_S_8_ mediators. (**f**) Galvanostatic charge−discharge curves for cathodes with and without the Mo_6_S_8_ mediator. (**g**) CV curves for cathodes with and without the Mo_6_S_8_ mediator. Reprinted with permission from Ref. [110]. Copyright 2019 American Chemical Society.

**Figure 9 nanomaterials-12-04341-f009:**
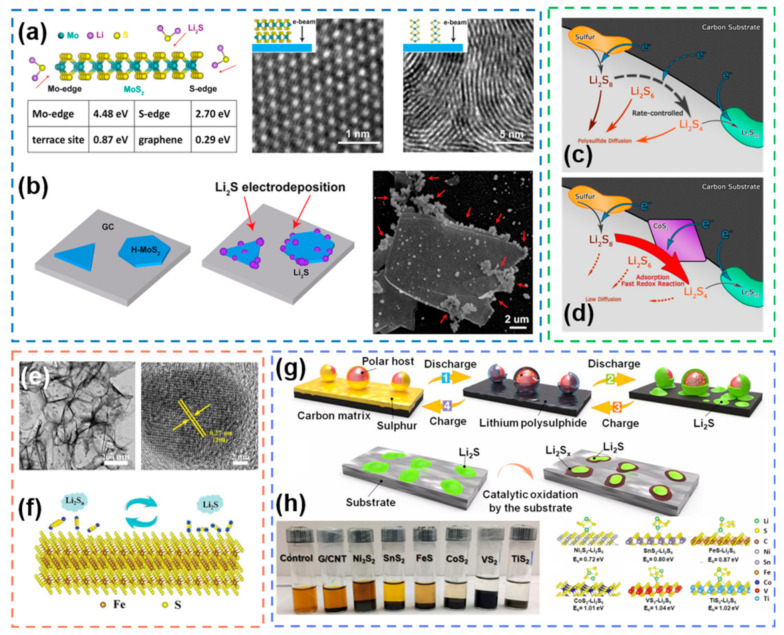
(**a**) Schematic illustration of the interaction between Li_2_S and different MoS_2_ atomic sites. (**b**) Schematics and SEM image of Li_2_S electrochemical deposition onto different sites of MoS_2_. Reprinted with permission from Ref. [113]. Copyright 2014 American Chemical Society. (**c**) Schematic illustration of rate-controlled reduction of polysulfides in pure carbon/sulfur cathodes. (**d**) Schematic illustration of carbon/sulfur cathode doped with CoS_2_ for accelerating polysulfide conversion. Reprinted with permission from Ref. [114]. Copyright 2016 American Chemical Society. (**e**) High-resolution transmission electron microscopy (HRTEM) images of FeS_2_−containing cathode. (**f**) Schematic diagram of FeS_2_ electrocatalytic sites for polysulfides. Reprinted from Ref. [115]. (**g**) Schematic diagram of Li_2_S catalytic oxidation on the surface of the material substrate. (**h**) Adsorption of Li_2_S_6_ by metal sulfides and corresponding simulations of surface adsorption. Reprinted from Ref. [116].

**Figure 10 nanomaterials-12-04341-f010:**
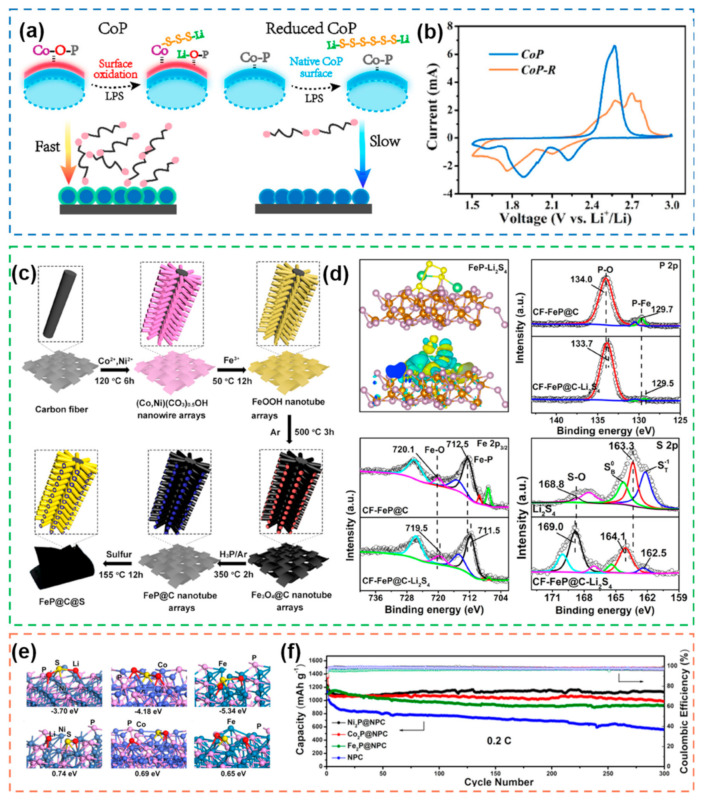
(**a**) Schematic diagram of the adsorption and catalytic process of polysulfides on the surface of CoP and the surface of reduced CoP. (**b**) CV curves of CoP cathode and reduced CoP cathode. Reprinted with permission from Ref. [119]. Copyright 2018 American Chemical Society. (**c**) Schematic diagram of cathode synthesis of carbon nanoarray−coated FeP. (**d**) Electron differential density and XPS spectra of Li_2_S_4_−FeP. Reprinted with permission from Ref. [120]. Copyright 2019 American Chemical Society. (**e**) Adsorption of various transition metal phosphide surfaces. (**f**) Long−cycle performance of various transition metal phosphide cathodes. Reprinted with permission from Ref. [121]. Copyright 2017 American Chemical Society.

**Figure 11 nanomaterials-12-04341-f011:**
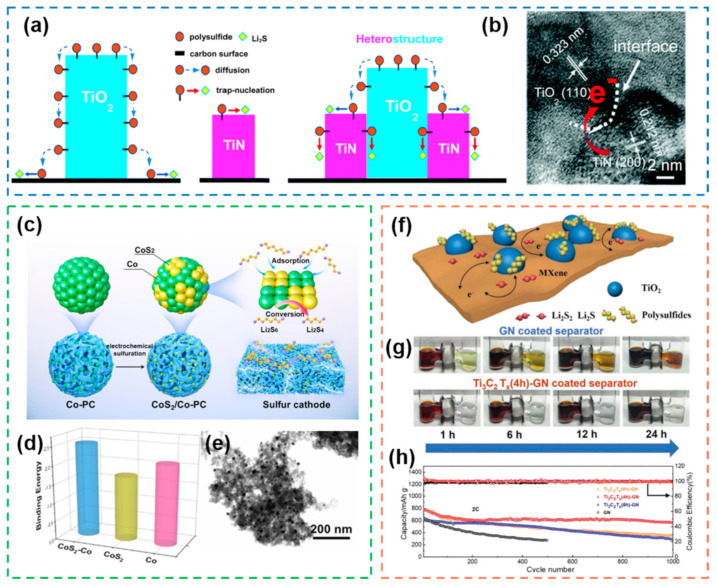
(**a**) Schematic of polysulfide catalytic processes on TiN, TiO_2_, and TiO_2_−TiN heterostructure surfaces. (**b**) HR−TEM images of the prepared TiO_2_−TiN heterostructure. Reprinted with permission from Ref. [128]. Copyright 2017 Royal Society of Chemistry. (**c**) Schematic diagram of the synthesis of checkerboard−like heterostructure by sulfonation reaction. (**d**) Binding energy of Co, CoS_2_ and CoS_2_−Co to Li_2_S_6_. (**e**) SEM image of the checkerboard−like nanostructure. Reprinted with permission from Ref. [129]. Copyright 2022 Elsevier. (**f**) Schematic diagram of adsorption and catalysis of polysulfides on Ti_3_C_2_T_x_/TiO_2_ heterostructure surface. (**g**) Suppression of the shuttle effect by heterostructure interlayer. (**h**) The cycling performance of Li−S batteries with heterostructure interlayers. Reprinted with permission from Ref. [133]. Copyright 2019 John Wiley and Sons.

**Figure 12 nanomaterials-12-04341-f012:**
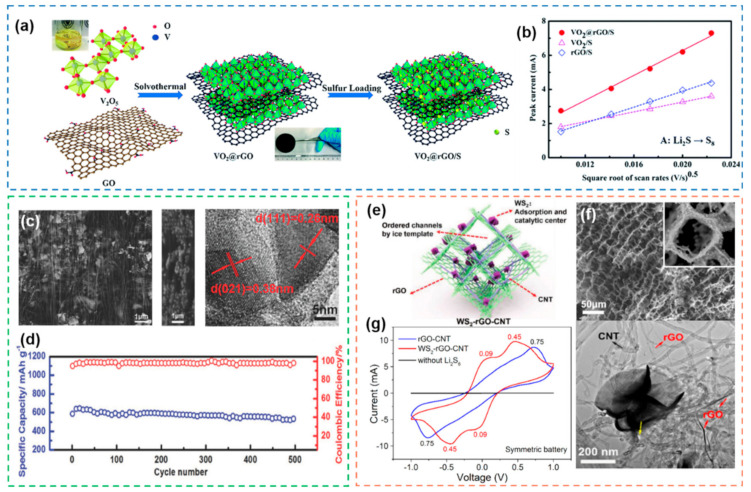
(**a**) Schematic of the synthesis of VO_2_@rGO/S cathode. (**b**) Diffusion coefficients of VO_2_@rGO/S, rGO/S, and VO_2_/S cathodes. Reprinted with permission from Ref. [137]. Copyright 2019 Royal Society of Chemistry. (**c**) SEM and HRTEM image of the MoP_2_/CNTs interlayer. (**d**) The cycling performance of Li−S batteries with MoP_2_/CNTs interlayers. Reprinted with permission from Ref. [138]. Copyright 2017 John Wiley and Sons. (**e**) Schematic diagram of 3D porous WS_2_−rGO−CNTs. (**f**) SEM and HRTEM image of ternary heterostructure. (**g**) CV curve of rGO−CNTs binary heterostructures and WS_2_−rGO−CNTs ternary heterostructure. Reprinted with permission from Ref. [139]. Copyright 2018 American Chemical Society.

**Figure 13 nanomaterials-12-04341-f013:**
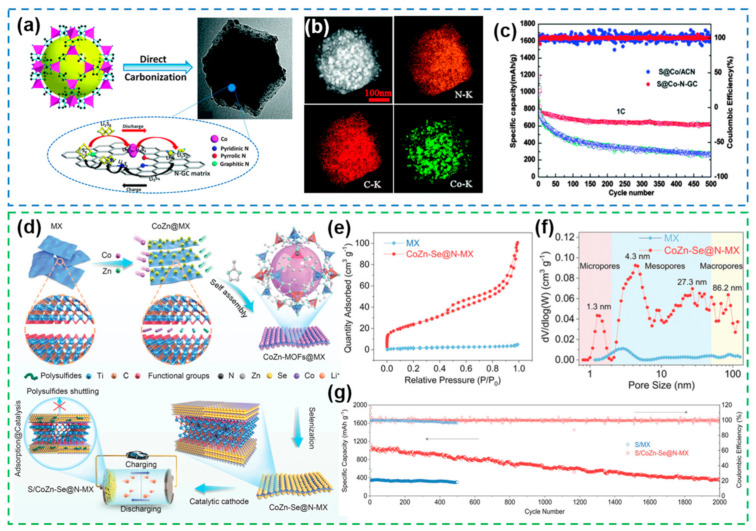
(**a**) Schematic of the interaction of Co−N−GC heterostructures with polysulfides. (**b**) The corresponding elemental mapping of Co−N−GC heterostructures. (**c**) Cycling performance comparisons between the S@Co−N−GC and the S@Co/ACN cathodes. Reprinted with permission from Ref. [147]. Copyright 2016 Royal Society of Chemistry. (**d**) Schematic illustration of the synthesis process of the CoZn−Se@N−MX heterostructure and its application in the electrocatalysis of Li−S batteries. (**e**) N_2_ adsorption−desorption isotherm curves of heterostructure. (**f**) The pore size distribution of heterostructure. (**g**) The cycling performance of Li−S batteries with CoZn−Se@N−MX heterostructure cathodes and S/MX cathodes. Reprinted with permission from Ref. [148]. Copyright 2021 John Wiley and Sons.

**Figure 14 nanomaterials-12-04341-f014:**
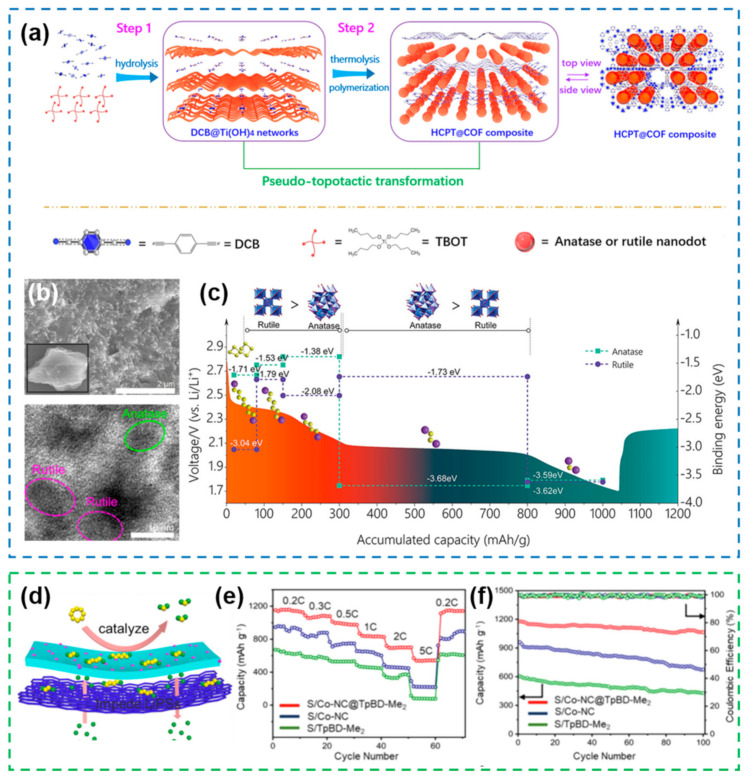
(**a**) Schematic illustration of the synthesis of hybrid−crystal−phase TiO_2_/COFs heterostructures. (**b**) SEM and TEM image of hybrid−crystal−phase TiO_2_/COFs heterostructures. (**c**) Schematic diagram of the binding energy of rutile and anatase for various polysulfides. Reprinted from Ref. [152]. (**d**) Schematic illustration of COFs/MOFs heterostructure catalyzed polysulfide conversion. (**e**) The rate performance of COFs/MOFs heterostructure and single component. (**f**) The cycling performance of COFs/MOFs heterostructure and single component. Reprinted with permission from Ref. [153]. Copyright 2022 Elsevier.

**Figure 15 nanomaterials-12-04341-f015:**
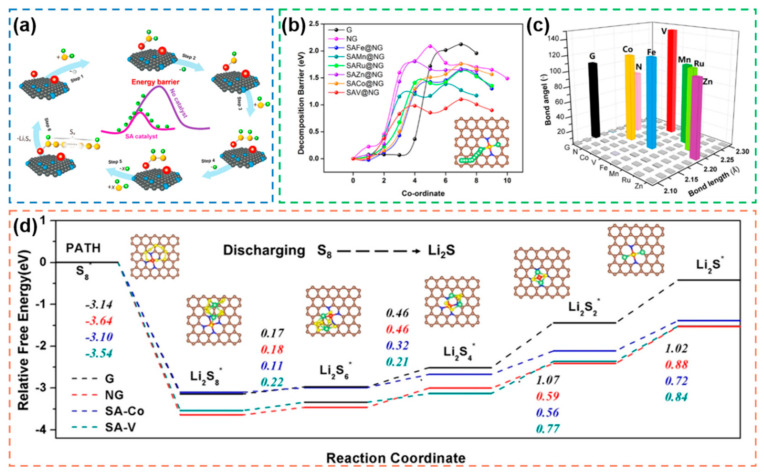
(**a**) Schematic of the mechanism for SAFe catalyzed Li_2_S delithiation reaction. Reprinted with permission from Ref. [169]. Copyright 2019 Elsevier. (**b**) Decomposition barriers of Li_2_S on different substrates including graphene, NG, SAFe@NG, SAMn@NG, SARu@NG, SAZn@NG, SACo@NG, and SAV@NG. (**c**) Bond angle (Li−S−Li) and bond length (Li−S) of Li_2_S on the graphene, NG, SACo@NG, SAV@NG, SAFe@NG, SAMn@NG, SARu@NG, and SAZn@NG. (**d**) Gibbs free energy profiles for the reduction of polysulfides on graphene, NG, SACo@NG, and SAV@NG. (The * represents the active substance (S_8_-Li_2_S) bound to different substrates.) Reprinted with permission from Ref. [62]. Copyright 2020 American Chemical Society.

**Figure 16 nanomaterials-12-04341-f016:**
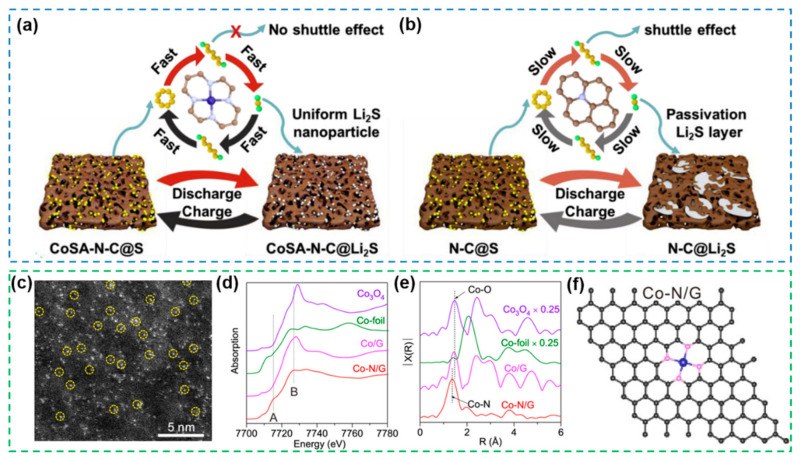
Schematic illustration of the effect of (**a**) CoSA-N-C and (**b**) N-C in improving the conversion kinetics of active materials, and mediating the deposition of Li_2_S. Reprinted with permission from Ref. [171]. Copyright 2020 Elsevier. (**c**) HAADF-STEM image of Co-N/G. (**d**) XANES and (**e**) FT-EXAFS in R space for Co-N/G and reference samples including Co/G, Co-foil, and Co_3_O_4_. (**f**) Structures of Co-N/G used in first-principles calculations. Reprinted with permission from Ref. [172]. Copyright 2019 American Chemical Society.

**Figure 17 nanomaterials-12-04341-f017:**
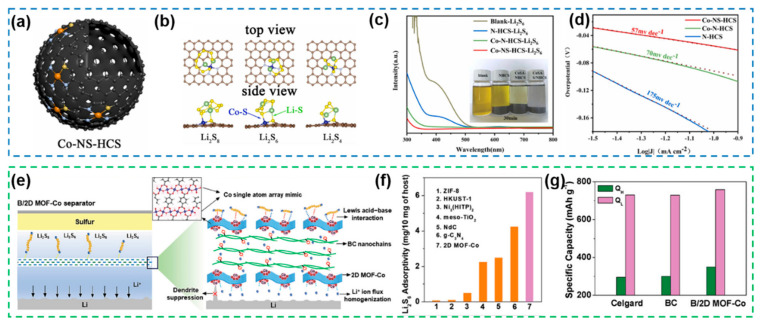
(**a**) Schematic illustration of Co−NS−HCS. (**b**) The calculated configurations of LiPS on Co−NS−HCS substrate with lowest system energies. (**c**) UV−vis spectra of Li_2_S_6_ solution with addition of different adsorbent after 30 min. (**d**) Tafel plots of Co−NS−HCS, Co−N−HCS and N−HCS. Reprinted with permission from Ref. [173]. Copyright 2022 Elsevier. (**e**) Schematic illustration for the Li−S batteries with B/2D MOF−Co separators. (**f**) Comparison of Li_2_S_6_ capture capacity for various reported materials. (**g**) Comparison of the high-discharge plateau (Q_H_) and low−discharge plateau (Q_L_) capacities for the Li−S batteries with different separators. Reprinted with permission from Ref. [174]. Copyright 2020 John Wiley and Sons.

**Figure 18 nanomaterials-12-04341-f018:**
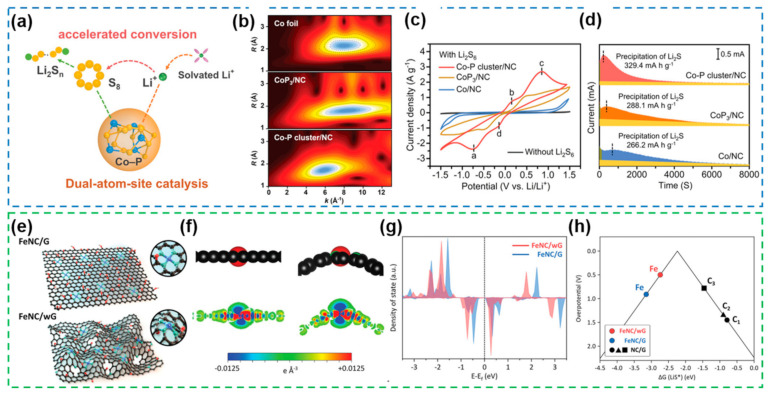
(**a**) Schematic illustration of the Co−P cluster/NC. (**b**) WT−EXAFS plots of the Co−P cluster/NC, CoP_3_/NC, and Co foil. (**c**) CV curves of Li_2_S_6_ symmetric batteries with different electrodes at a scan rate of 3 mV s^−1^. (**d**) Potentiostatic discharge profiles at 2.05 V on different electrodes with Li_2_S_8_ catholyte for evaluating the nucleation kinetics of Li_2_S. Reprinted with permission from Ref. [175]. Copyright 2022 John Wiley and Sons. (**e**) Illustration of FeNx sites on planar graphene (FeNC/G) and wG (FeNC/wG). (**f**) The structures of FeNC/G and FeNC/wG simulated in DFT calculations. (**g**) Density of states projected onto the d orbital of Fe atom for FeNC/G and FeNC/wG. (**h**) Volcano plot of the overpotential for the last step of the sulfur reduction reaction. Reprinted with permission from Ref. [176]. Copyright 2022 John Wiley and Sons.

**Figure 19 nanomaterials-12-04341-f019:**
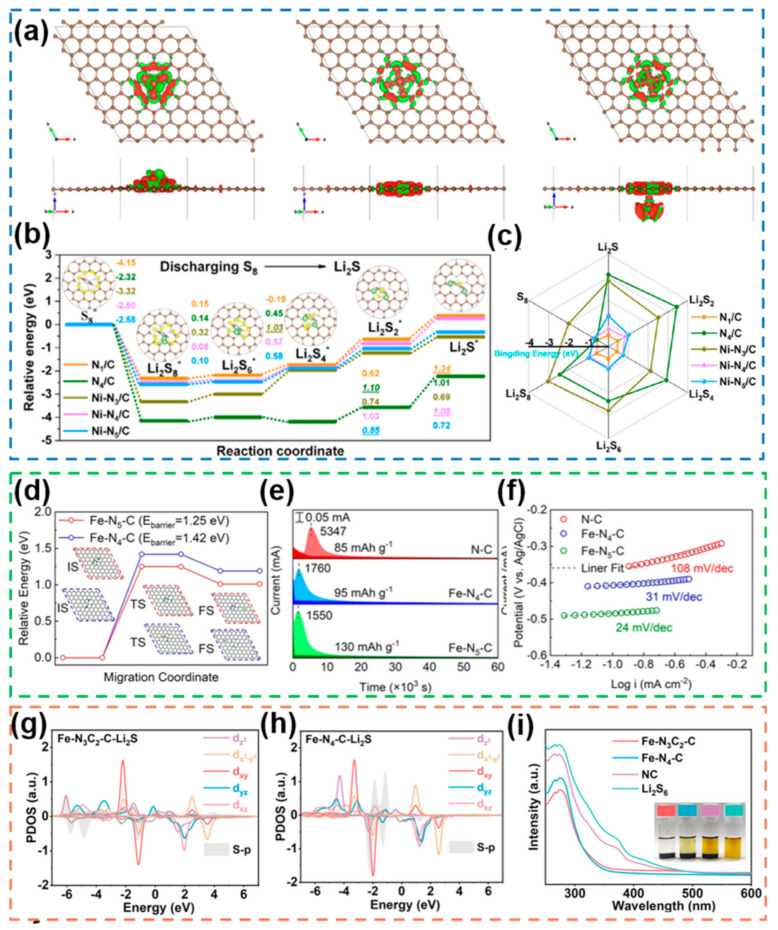
(**a**) The 3D isosurfaces of charge density difference for Ni−N_3_/C, Ni−N_4_/C, and Ni−N_5_/C. (**b**) Relative energy profiles of the discharge process for different catalyst models. (**c**) Binding energies of LiPSs on the five catalyst models. Reprinted with permission from Ref. [177]. Copyright 2021 American Chemical Society. (**d**) Energy profiles of Li_2_S decomposition. (**e**) Li_2_S precipitation profiles. (**f**) Tafel plots of Li_2_S oxidization of N−C, Fe−N_4_−C, and Fe−N_5_−C. Reprinted with permission from Ref. [178]. Copyright 2021 John Wiley and Sons. PDOS of (**g**) Fe−N_3_C_2_−C and (**h**) Fe−N_4_−C after adsorption of Li_2_S. (**i**) UV−vis spectra of Li_2_S_6_ solutions with different samples. Reprinted with permission from Ref. [179]. Copyright 2021 American Chemical Society.

**Figure 20 nanomaterials-12-04341-f020:**
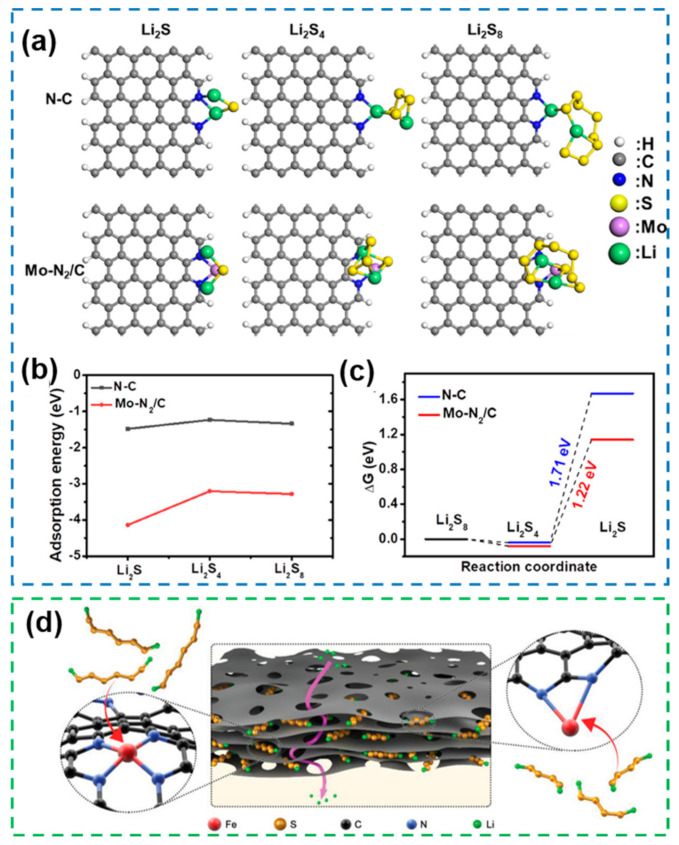
(**a**) The optimized adsorption conformations of LiPS on N−C and Mo−N_2_/C. (**b**) Adsorption energies of Mo−N_2_/C and N−C for Li_2_S_8_, Li_2_S_4_, and Li_2_S. (**c**) Gibbs free energy changes in Li_2_S_8_, Li_2_S_4_, and Li_2_S on Mo−N_2_/C and N−C. Reprinted with permission from Ref. [180]. Copyright 2020 American Chemical Society. (**d**) Schematic illustration of the state of polysulfide constrained in layered carbon and the ion diffusion pathways provided by the porous structure. Reprinted with permission from Ref. [181]. Copyright 2018 John Wiley and Sons.

**Figure 21 nanomaterials-12-04341-f021:**
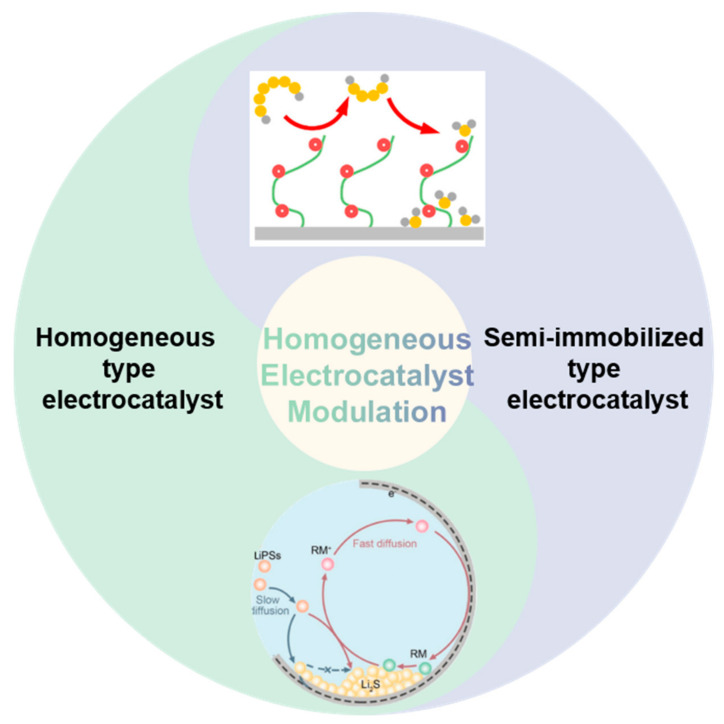
Schematic illustration of two modulation strategies for homogeneous electrocatalyst. Reprinted with permission from Ref. [182]. Copyright 2021 American Chemical Society.

**Figure 22 nanomaterials-12-04341-f022:**
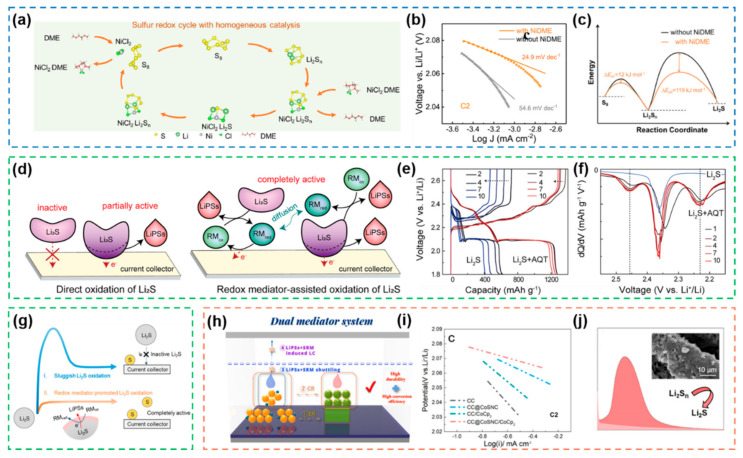
(**a**) Schematic diagram of a LiPS conversion cycle with recyclable NiDME additives. (**b**) Tafel plots of the C2 peaks of the CV profiles of Li−S batteries with and without NiDME. (**c**) The activation energies (Ea) of the discharge processes. Reprinted with permission from Ref. [66]. Copyright 2020 Elsevier. (**d**) Schematic illustration of Li_2_S oxidation reaction with and without RM participation. (**e**) GCD profiles of Li_2_S electrode cycled with and without AQT (80 mM) participation. (**f**) Corresponding dQ/dV curves at 0.1 C. Reprinted with permission from Ref. [183]. Copyright 2019 Elsevier. (**g**) Schematic first charge profiles of ASSLSBs with and without AQT. Reprinted with permission from Ref. [184]. Copyright 2021 American Chemical Society. (**h**) Schematic diagram of LiPSs conversion reactions in dual−mediator system. (**i**) Tafel plots of the asymmetrical cells without and with different redox mediators. (**j**) Potentiostatic discharge profile of a Li_2_S_8_ solution at 2.05 V for the cell with CC@ CoSNC/CoCp_2_. The insets are SEM images showing the corresponding nucleation of Li_2_S. Reprinted with permission from Ref. [185]. Copyright 2022 American Chemical Society.

**Figure 23 nanomaterials-12-04341-f023:**
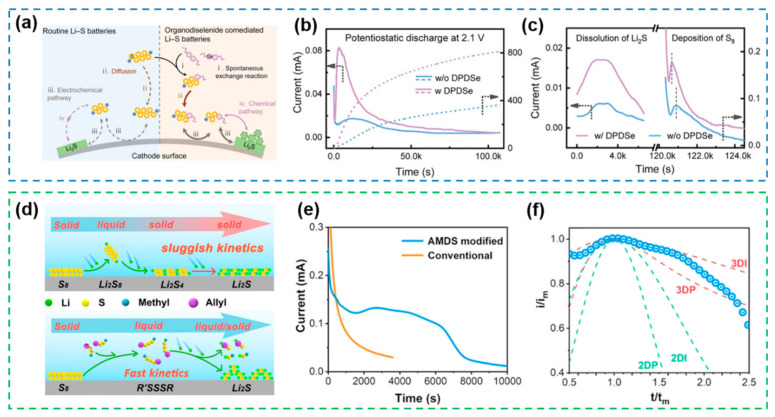
(**a**) Schematic of routine and organodiselenide-comediated reaction pathway for Li–S batteries. (**b**) Chronoamperometry curves at 2.1 V showing the kinetics of Li_2_S deposition. (**c**) Enlarged PITT profiles for Li_2_S dissolution and S_8_ deposition. Reprinted with permission from Ref. [186]. Copyright 2021 John Wiley and Sons. (**d**) Schematic illustrations of the phase conversion in Li-S batteries with and without AMDS. (**e**) Current–time transients curves obtained at 2.08 V and (**f**) Corresponding electrochemical nucleation model simulation diagram. Reprinted with permission from Ref. [187]. Copyright 2021 American Chemical Society.

**Figure 24 nanomaterials-12-04341-f024:**
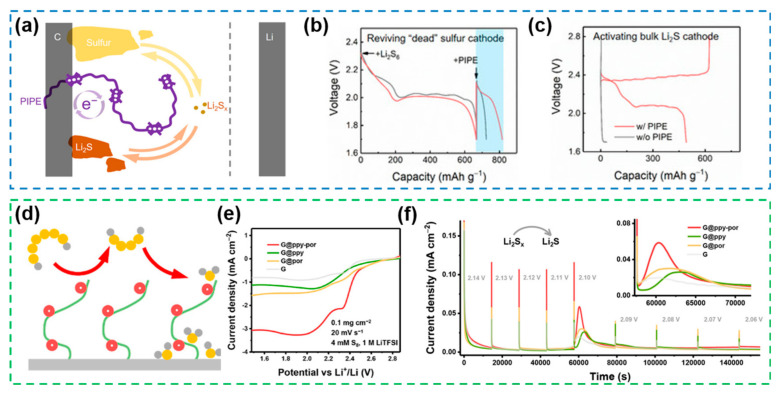
(**a**) Schematic diagram of the S−Li_2_S_x_−Li_2_S reversible conversion process with RM participation. (**b**) The reviving test of batteries discharged to 1.7 V. (**c**) GCD curves of Li_2_S cathode batteries with/without PIPE. Reprinted with permission from Ref. [188]. Copyright 2020 John Wiley ad Sons. (**d**) Schematic illustration of the semi−immobilized electrocatalysts with simultaneous heterogeneous and homogeneous electrocatalytic functions. (**e**) LSV curves for S_8_ reduction tested in a 4 mM S_8_ ether−based electrolyte. (**f**) PITT curves for the Li_2_S deposition process of Li−S batteries without and with different electrocatalysts. Reprinted with permission from Ref. [182]. Copyright 2021 American Chemical Society.

**Figure 25 nanomaterials-12-04341-f025:**
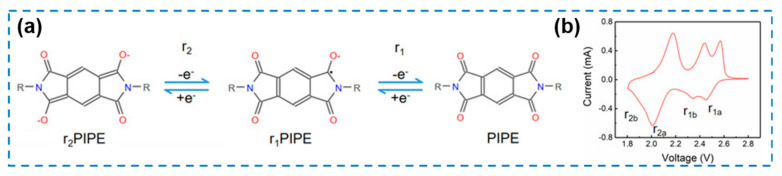
(**a**) Illustration of the two−electron redox mechanism of conjugated imide segments in PIPE. (**b**) CV curves of PIPE. Reprinted with permission from Ref. [188]. Copyright 2020 John Wiley and Sons.

**Figure 26 nanomaterials-12-04341-f026:**
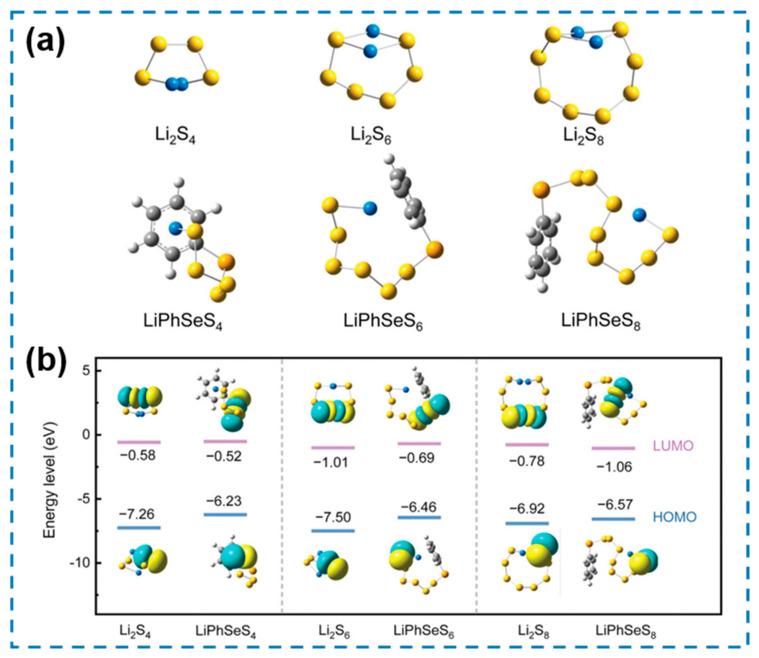
Simulation results of LiPSs and LiPhSePSs. (**a**) Optimized molecular structures and (**b**) LUMO and HOMO energy levels of different Li_2_Sn and LiPhSeS_n_. Reprinted with permission from Ref. [186]. Copyright 2021 John Wiley and Sons.

**Table 1 nanomaterials-12-04341-t001:** A summary of heterogeneous electrocatalyst for Li-S batteries with different dimensions.

Heterogeneous Electrocatalyst	Material	Nanostructure	Cycle Performance [mAh g^−1^]	Ref.
Metal compound electrocatalyst	Titanium-deficient anatase TiO_2_	0D	511/2 C/1000th	[89]
	Fe_3_O_4_ nanocrystals/CNTs	0D/1D	538.5/1 C/1800th	[83]
	Pt NPs/graphene	0D/2D	789/0.1 C/100th	[57]
	CNTs/V_2_O_5_ nanoplates	1D/2D	887.2/1 C/500th	[95]
	Fe_2_O_3_ NPs/Porous graphene	0D/3D	516/5 C/1000th	[77]
	FeS_2_ nanoclusters/Porous carbon	0D/3D	1171/0.4 C/100th	[115]
	FeP/Carbon nanoarray	1D	500/1 C/200th	[120]
	MnO_2_ nanosheets	2D/	245/2 C/2000th	[86]
	Graphene-like Co_9_S_8_	2D	643/2 C/400th	[109]
	Layered MoS_2_	2D	over 800/0.2 C/300th	[113]
	VN nanoribbon/Graphene	3D	917/1 C/200th	[100]
	Mesoporous TiN	3D	644/0.5 C/500th	[101]
	MIL-88A	3D	200/0.5 C/1000th	[122]
	Nanoporous Co and N-Codoped Carbon Composite	3D	477/1 C/1000th	[123]
Heterostructure electrocatalyst	TiO_2_-TiN	0D/0D	704/1 C/2000th	[128]
	CoS_2_-Co	0D/0D	1227/0.2 C/150th	[129]
	MoP_2_ NPs/CNTs	0D/2D	543/1 C/500th	[138]
	TiO_2_/Ti_3_C_2_T_x_	0D/2D	576/2 C/1000th	[133]
	CoZn-Se NPs/MXene	0D/2D	849/3 C/300th	[148]
	TiO_2_ nanodots/COFs	0D/2D	875/0.5 C/800th	[152]
	Nitrogen-doped MXene/nitrogen-doped carbon nanosheet-nickel	0D/2D	588/1 C/500th	[140]
	CuCo_2_S_4_ NPs/CNTs	0D/2D	627/1 C/500th	[135]
	Co NPs/N-doped graphitic carbon	0D/3D	625/1 C/500th	[147]
	COFs/MOFs	2D/3D	636.2/1 C/500th	[153]
	WS_2_-rGO-CNTs	0D/1D/2D (3D)	1227/0.1 C/100th	[139]
	CuZnS QDs-decorated (NiCo)–S@rGO/oxdCNT	0D/1D/2D (3D)	537/0.1 C/100th	[134]
Single-atom electrocatalyst	SAV/Graphene	0D/2D	551/0.5 C/400th	[62]
	CoSA/Carbon nanosheet	0D/2D	675/1 C/1000th	[171]
	Mo-N_2_/Cabon nanosheet	0D/2D	817/2 C/550th	[180]
	SAFe/N-porous carbon	0D/3D	427/0.1 C/300th	[166]
	SACo/Hollow carbon spheres	0D/3D	692/1 C/500th	[173]
	Co-P cluster/Carbon matrix	0D/3D	746/1 C/1000th	[175]

**Table 2 nanomaterials-12-04341-t002:** A summary of heterogeneous electrocatalyst for the cathode of Li-S batteries.

Heterogeneous Electrocatalyst	Material	Sulfur Content of Cathode [%]	Catalyst Content of Cathode [%]	Specific Surface Area [m^2^ g^−1^]	Ref.
Metal compound electrocatalyst	Titanium-deficient anatase TiO_2_	75	5.75	165	[89]
	Fe_3_O_4_ nanocrystals/CNTs	60	4.52	301.2	[83]
	Pt NPs/graphene	83	4.25	405	[57]
	CNTs/V_2_O_5_ nanoplates	75	25	108	[95]
	Fe_2_O_3_ NPs/Porous graphene	56	17.6	37	[77]
Heterostructure electrocatalyst	CoS_2_-Co	76	2.76	1232	[129]
	WS_2_-rGO-CNTs	55	21.1	315	[139]
Single-atom electrocatalyst	SAV/Graphene	80	0.86	781.9	[62]
	SACo/Graphene	80	0.77	841.8	[62]
	Mo-N_2_/Cabon nanosheet	81	0.38	1038	[180]
	Co-P cluster/Carbon matrix	70	0.3	339	[175]

**Table 3 nanomaterials-12-04341-t003:** A summary of homogeneous electrocatalyst for Li-S batteries with different functional groups/structures.

Homogeneous Electrocatalyst	Material	Functional Group/Structure	Cycle Performance [mAh g^−1^]	Ref.
Homogeneous-type electrocatalyst	Nickel dimethoxyethane chloride adduct	NiCl_2_	784/1 C/500th	[66]
	Anthraquinone derivative 1,5-bis(2-(2-(2-methoxyethoxy) ethoxy) ethoxy) an-thra-9,10-quinone	anthraquinone	850/1 C/500th	[183]
	CoSNC (CoS_1.097_ nanoparticles (NPs) embedded in nitrogen-doped porous carbon sheets) with cobaltocene	transition metal metallocenes	509/2 C/1200th	[185]
	Diphenyl diselenide (DPDSe)	phenylselenides	720/0.5 C/350th	[186]
	Allyl methyl disulfide(AMDS)	disulfide structure	798/0.8 C/50th	[187]
	Ethyl viologen diperchlorate (EV(ClO_4_)_2_)	ethyl viologen structure	395/1.0 mA cm^−2^/500th	[191]
Semi-immobilized-type electrocatalyst	PIPE	imide structure	916/0.5 C/300th	[188]
	G@ppy-por	porphyrin structure	904/2 C/50th	[182]
	BPI	imide structure	560/0.5 C/100th	[189]
	CoCp_2_	transition metal metallocenes	553/2 C/100th	[190]

**Table 4 nanomaterials-12-04341-t004:** Comparison of the advantages and disadvantages of heterogeneous and homogeneous electrocatalysts and the criteria for judging the process/mechanism of electrocatalysts.

Types of Electrocatalysts	Advantages	Disadvantages	Process/Mechanism Judgment Basis
Heterogeneous electrocatalyst	1. Convenient and efficient electrocatalytic mechanism 2. Catalytic performance conveniently modulated by nanostructures 3. Improved anode structure and performance	1. Decreased activity as the active surface be gradually covered 2. As an inactive component affects battery energy density	Judgment in the balance between electrocatalytic performance and catalytic activity retention based on performance requirements (heterogeneous electrocatalysts usually provide more direct electrochemical performance enhancement but less stability, while homogeneous electrocatalysts provide more stable electrocatalytic activity)
Homogeneous electrocatalyst	1. Enhanced kinetics in terms of reaction mechanism 2. Active site enrichment (full exposure to active substances)	1. Decreased activity with the shuttle effect 2. Complex catalytic mechanism	

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
