# Peer review of "Advanced Nanostructured Materials for Electrocatalysis in Lithium–Sulfur Batteries"

_nanomaterials, 2022, doi:10.3390/nano12234341_

Round 1

Reviewer 1 Report

The manuscript titled ‘Advanced nanostructured materials for electrocatalysis in lithium-sulfur batteries’ documents the recent progress in electrocatalyst employed in Li – S battery. The content of the manuscript is concise and easy to understand. However, the authors need to correct the following points:

1. Rewrite the sentence on lines 186-188.

2. Please rewrite the manuscript and avoid using the author's name to cite research results.

3. Table 1: catalyst mass, S/Li2S mass used in the electrode should be added so that their relationship with durability and battery capacity can be easily visualized.

4. The surface area of the catalyst is an important parameter that determines the rate of the catalytic reaction and the adsorption of polysulfide on the surface of the catalyst, so the author should add a table showing these parameters and detailed discussion.

Author Response

The manuscript titled ‘Advanced nanostructured materials for electrocatalysis in lithium-sulfur batteries’ documents the recent progress in electrocatalyst employed in Li-S battery. The content of the manuscript is concise and easy to understand. However, the authors need to correct the following points:

  1. Rewrite the sentence on lines 186-188.

RESPONSE: Thank you for your suggestion. We have modified this sentence in the manuscript to make it clearer:

“Zero-dimensional (0D) ultrafine Fe3O4 nanocrystals were introduced in situ into carbon nanospheres as electrocatalysts to modify the separator, which can effectively enhance the electrochemical performance of Li-S batteries.” (Lines 193~196)

  1. Please rewrite the manuscript and avoid using the author's name to cite research results.

RESPONSE: Thank you very much for your reminder. As you mentioned, it is inappropriate to use the author's name to cite the research results. We have corrected all the phrases in the text to ensure the overall soundness of the article:

“Inspired by this, traditional electrocatalyst Pt nanoparticles (NPs) are loaded on graphene as the cathode host of Li-S batteries (Fig. 3a), and found that its specific capacity was about 40% higher than that of conventional graphene/sulfur composite cathodes (Fig. 3b).” (Lines 153~156)

“The modification of Au NPs on commercial acetylene black/sulfur composite cathodes (CB-S-Au) is also capable of electrocatalysis. Rapid conversion of polysulfides is achieved by regulating the chemistry between Au NPs and polysulfides, thus improving the performance of Li-S batteries (Fig. 3c, d).” (Lines 157~161)

“Loading α-Fe2O3 NPs on three-dimensional (3D) porous graphene (Fig. 4a) as the cathode of Li-S batteries can achieve electrocatalytic effects.” (Lines 179~181)

“The introduction of Fe3O4-encapsulated NPs into 1D carbon nanotubes (CNTs) as hosts for Li-S batteries can effectively accelerate the electrochemical reactions inside the batteries.” (Lines 186~188)

“Zero-dimensional (0D) ultrafine Fe3O4 nanocrystals were introduced in situ into carbon nanospheres as electrocatalysts to modify the separator, which can effectively enhance the electrochemical performance of Li-S batteries.” (Lines 193~196)

“Two-dimensional (2D) ultrathin MnO2 nanosheets were shown to accelerate the conversion of polysulfides to insoluble Li2S/Li2S2.” (Lines 210~211)

“Titanium-deficient anatase TiO2 (TDAT) as a separator modification material can also improve the electrochemical performance of Li-S batteries.” (Lines 220~222)

“Besides TiO2, nano-Magnéli-phase titanium oxide (TinO2n-1) also occupies an important position in the electrocatalysis of Li-S batteries. During the electrochemical reaction, TinO2n-1 is capable of bonding or electrocatalysis with sulfur, which has a significant improvement on the battery performance.” (Lines 226~229)

“The composite of VN nanoribbons with porous graphene and used as cathode materials for Li-S batteries can effectively accelerated the reaction kinetics inside the batteries.” (Lines 225~257)

“Mesoporous TiN with high specific surface area synthesized by solid-solid phase separation method used zinc titanate as raw material exhibited good electrochemical properties.” (Lines 270~272)

“Therefore, the introduction of WN with different nanostructures into various types of carbon materials can exhibit good electrocatalytic properties.” (Lines 277~279)

“A Co9S8 with interconnected graphene-like nanostructures was synthesized and demonstrated to interact strongly with polysulfides when used as a cathode.” (Lines 302~303)

“For example, Mo6S8 can be chelated with polysulfides to form LixMo6S8.” (Lines 311~312)

“Therefore, there is a difference in the properties between the unsaturated-coordinated edge metal sites and the saturated-coordinated surface metal sites in MoS2.” (Lines 333~335)

“In addition, CoS2 was also added as an electrocatalyst to the carbon/sulfur cathode to improve the performance of Li-S batteries by accelerating the redox kinetics of polysulfides.” (Lines 337~339)

“FeS2 is also used as an electrocatalyst in Li-S batteries. Monodisperse sub-10 nm FeS2 nanoclusters were modified on porous carbon surfaces for enhanced electrochemical kinetics.” (Lines 342~344)

“To further explore the potential of metal sulfides for electrocatalysis in Li-S batteries, the ability of various metal sulfides to adsorb polysulfides and oxidize Li2S was systematically explored.” (Lines 345~347)

“In 2018, Co-O-P was found to be formed in surface oxidized CoP, which enables Co sites to interact strongly with polysulfides and generate Co-S bonds.” (Lines 370~372)

“In addition, the special structure of the carbon nanoarray-coated FeP materials can facilitates the fast conduction of ions/electrons.” (Lines 375~376)

“Due to the complex electrocatalytic properties, it is necessary to explore in depth the electrocatalytic effect of various types of transition metal phosphides on Li-S batteries.” (Lines 379~381)

“In response to this problem, the concept of bifunctional heterogeneous structure was proposed. The combination of metal oxides with high adsorption and metal nitrides with high conductivity enables the effective preparation of heterogeneous structured electrocatalysts.” (Lines 414~418)

“Based on this idea, a checkerboard-like heterostructure electrocatalyst was designed by the sulfonation reaction.” (Lines 421~422)

“A Ti3C2Tx/TiO2 heterostructure catalysts was successfully prepared by in situ oxidation and used as interlayers.” (Lines 428~430)

“Aiming at the problem of good adsorption but poor conductivity of VO2, in situ growth of VO2 on the surface of 2D reduced graphene oxide (rGO) by solvothermal method to construct heterostructures is a good solution.” (Lines 451~453)

“Aiming at the problem of good adsorption but poor conductivity of VO2, in situ growth of VO2 on the surface of 2D reduced graphene oxide (rGO) by solvothermal method to construct heterostructures is a good solution.” (Lines 153~156)

“A heterostructure composed of CNTs and MoP2 NPs was combined and applied to the interlayer of Li-S batteries.” (Lines 457~458)

“Carbonization of MOF precursors is a common method to prepare MOF-based hetero-structures. For example, 3D porous cobalt-N-doped graphitic carbon (Co-N-GC) hetero-structures can be obtained by carbonizing Co-MOF.” (Lines 489~492)

“Based on the interaction between MOF and MXene, bimetallic selenides on nitro-gen-doped MXene (CoZn-Se@N-MX) enable successful self-assembly.” (Lines 496~498)

“Homogeneous embedding of anatase/rutile TiO2 nanodots (10 nm) into porous COFs enables to obtain a hybrid-crystal-phase TiO2/COFs heterostructure.” (Lines 519~521)

“In addition, calculations were able to demonstrate that the binding energy of different phases of TiO2 on polysulfides were different.” (Lines 523~524)

“Wrapping the porous MOFs with microporous COFs enables to obtain heterogeneous structures with advanced nanostructures, where the inner MOFs act as catalysts to accelerate the conversion of polysulfides, while the outer COFs layer restricts the shuttle of polysulfides through polar functional groups.” (Lines 528~532)

“For this reason, the interaction between isolated atoms and supports was proposed in 1978, through which the movement and aggregation of isolated atoms can be prevented and a uniform and stable distribution of active sites can be constructed, thus positively affecting the selectivity, stability and electrocatalytic activity of the catalysts.” (Lines 564~568)

“SACs with saturated coordination configurations were first applied to Li-S batteries in 2018, when single-atom Fe (SAFe) coordinated with four pyrrole N on the surface of the supports was used as an electrocatalyst to improve the rate performance and cycling life-time of the battery, demonstrating the electrocatalytic effect of SAFe on the redox reaction of LiPSs. To investigate the intrinsic mechanism of SACs in the electrochemical reaction of Li-S batteries, a novel theoretical simulation study was proposed, which demonstrated that SAFe can coordinate with Li2S thus weakening the Li-S bond, promoting the decomposition of Li2S and the growth of LiPSs chains, and that it can regenerate by gaining electrons during the charging phase (Fig. 15a).” (Lines 578~587)

“Therefore, based on the purpose of screening the most suitable metal atoms for the preparation of SACs for catalytic Li-S electrocatalysis, a novel theoretical model verified the effect of SACs constructed with different monometallic atoms in saturated coordination configurations on the electrocatalytic capacity of Li-S batteries.” (Lines 596~600)

“To deeply dissect the microstructure, the homogeneous dispersion state of Co on the support was analyzed by aberration-corrected scanning transmission electron microscopy (STEM) high-angle annular dark field (HAADF) (Fig. 16c) imaging, x-ray absorption near edge structure (XANES) (Fig. 16d) and extended x-ray absorption fine structure (EXAFS) (Fig. 16e), which elucidate the Co-N4 saturated coordination conformation of such SACs at the atomic level (Fig. 16f).” (Lines 632~638)

“The SACs constructed by co-coordination of N, S atoms also exhibited significant superiorities in accelerating the Li-S electrochemical reactions, for example, single-atom catalysts with Co-N3S-C coordination structures prepared by thiourea (Fig. 17a).” (Lines 648~650)

“With the purpose of simultaneously solving the problems of lithium dendrites and shuttle effects inherent to Li-S batteries, single-atom array mimics with Co-O4 coordination structures constructed from periodically arranged Co and O atoms on the surface of ultrathin metal-organic frameworks were successfully applied to Li-S batteries (Figure 17e).” (Lines 658~661)

“Based on the work of SACs, a dual-atom catalyst Co-P cluster/NC using P atoms as ligand atoms as well as catalytic progenitors (Fig. 18a) was proposed, and the local structure and chemical environment of Co atoms were further analyzed by XANES and WT-EXAFS (Fig. 18b), which demonstrated the Co-P cluster/NC configuration constructed by coordination of atomic Co with P, N.” (Lines 681~685)

“The introduction of highly folded structures into the support can modify the local coordination structure of the FeN4 group (Fig. 18e) thus achieving a simultaneous enhancement of the electrocatalytic activity of SACs at different current densities.” (Lines 689~692)

“To explore the effect of the coordination number of atoms on the catalytic activity of SACs, the investigators demonstrated the stronger adsorption capacity and facilitated conversion kinetics of LiPSs for the oversaturated coordination configuration of Ni-N5 by a com-bination of DFT calculations, electrochemical tests, and spectroscopic characterization.” (Lines 715~719)

“Fe-N5-type SACs in supersaturated coordination configuration demonstrated that Fe monoatoms with high N coordination number can strengthen the immobilization of LiPSs compared to saturated coordination (Fig. 19d) and effectively reduce the energy barriers for the conversion of LiPSs and Li2S nucleation (Fig. 19e, f). In order to address the problem of insufficient adsorption capacity of M-N4 configuration SACs for polysulfides, Fe-N3C2-C SACs constructed by using C and N as co-coordinating atoms were designed and prepared with asymmetric supersaturated coordination configurations, which not only form additional π-bonds with the p orbital hybridization of S to anchor LiPSs, but also effectively enhance the redox kinetics of LiPSs.” (Lines 724~732)

“Single-atom catalysts with Mo-N2-C conformation (Fig. 20a) with good adsorption of LiPSs and Li2S (Fig. 20b) and improved conversion kinetics (Fig. 20c) were applied to obtain Li-S batteries with high reversible capacity (743.9 mAh g-1 at 5C rate) and long cycling lifetime (after 550 cycles with a capacity decay of only 0.018% per cycle). Through a combination of supporting structure design and coordination environment modulation, SACs with FeN2 unsaturated coordination structures were introduced onto graphene with a pore-rich structure (Figure 20d), achieving uniform Li transport and effective adsorption of LiPSs.” (Lines 744~752)

“Nickel dimethoxyethane chloride adduct (NiDME) were selected as homogeneous catalysts for accelerating the redox conversion of sulfur as well as for inhibiting the shuttle effect of polysulfides.” (Lines 844~846)

“Anthraquinone derivative 1,5-bis(2-(2-(2-methoxyethoxy) ethoxy) ethoxy) an-thra-9,10-quinone (AQT) were incorporated into the electrolyte, which effectively enhanced the utilization of Li2S and prevented the deposition of "dead" Li2S.” (Lines 863~865)

“Further, AQT were applied to all-solid-state Li-S batteries, using its fast electron transfer pathways constructed between active materials and current collector to greatly reduce the dissociation barrier of Li2S (Fig. 22g), thus achieving fast Li-S redox reaction kinetics in a solid-state battery.” (Lines 876~879)

“To suppress the loss/deactivation of catalyst, the heterogeneous electrocatalyst CoSNC (CoS1.097 nanoparticles embedded in nitrogen-doped porous carbon sheets) were rationally coupled with the homogeneous electrocatalyst CoCp2 (cobaltocene), and their synergy was utilized to catalyze the electrochemical reaction of polysulfides (Fig. 22h, i).” (Lines 881~885)

“Based on this consideration, a diphenyl diselenide (DPDSe) was introduced into electrolyte, which could spontaneously react with the polysulfide to generate intermediate lithium phenyl selenium polysulfides (LiPhSePSs) with improved redox-mediated ability (Fig. 23a), thus accelerating the polysulfide bidirectional redox kinetics in reaction mechanism.” (Lines 905~909)

“Along the line of changing reaction mechanism, homogeneous electrocatalyst allyl methyl disulfide (AMDS)-mediated all-liquid-phase mechanism were further applied to Li-S batteries over a wide temperature range (room temperature to Cryogenic temperature).” (Lines 928~931)

“Starting from the design described above, a semi-immobilization strategy for the co-polymerization of small molecule imides with medium chain polyether to form space-constrained but catalytically active semi-immobilized RMs (PIPEs) was firstly pro-posed.” (Lines 954~956)

“Based on the semi-immobilization strategy, the semi-immobilized electrocatalyst were further extended to heterogeneous electrocatalysis, which verified the feasibility of coupling homogeneous electrocatalysis with non-homogeneous electrocatalysis.” (Lines 980~982)

  1. Table 1: catalyst mass, S/Li2S mass used in the electrode should be added so that their relationship with durability and battery capacity can be easily visualized.
  2. The surface area of the catalyst is an important parameter that determines the rate of the catalytic reaction and the adsorption of polysulfide on the surface of the catalyst, so the author should add a table showing these parameters and detailed discussion.

RESPONSE: Thank you for your suggestion. Your suggestion is very necessary and feasible. Here, we have additionally supplemented Table 2 to highlight the relevant parameters concerning the catalysis of cathode materials. The sulfur content of the cathode, the catalyst content of the cathode, and the specific surface area of its nanostructures were quantified and compared in detail:

Table 2. A summary of heterogeneous electrocatalyst for the cathode of Li-S batteries.

Heterogeneous electrocatalyst

Material

Sulfur content of cathode [%]

Catalyst content of cathode [%]

Specific surface area [m2 g-1]

Ref.

Metal compound electrocatalyst

Titanium-deficient anatase TiO2

75

5.75

165

[89]

Fe3O4 nanocrystals/CNTs

60

4.52

301.2

[83]

Pt NPs/graphene

83

4.25

405

[57]

CNTs /V2O5 nanoplates

75

25

108

[95]

Fe2O3 NPs/ Porous graphene

56

17.6

37

[77]

Heterostructures electrocatalyst

CoS2-Co

76

2.76

1232

[129]

WS2-rGO-CNTs

55

21.1

315

[139]

Single atoms electrocatalyst

SAV/ Graphene

80

0.86

781.9

[62]

SACo/ Graphene

80

0.77

841.8

[62]

Mo-N2/Cabon nanosheet

81

0.38

1038

[180]

Co-P cluster/Carbon matrix

70

0.3

339

[175]

“In particular, the electrocatalytic properties of typical cathode materials should be explored in more detail. Here, the sulfur content of the cathode, the electrocatalyst content of the cathode and the specific surface area are listed to facilitate a clearer comparison of the nanostructures of various types of non-homogeneous catalysts in relation to the capacity of the battery. It can be seen that single-atom electrocatalysts usually have a greater advantage in terms of specific surface area and catalyst usage compared to the other two types of electrocatalysts. Because single-atom electrocatalysts expose more active sites, their specific surface area tends to be larger, and their usage in the cathode is lower, enabling them to catalyze electrochemical reactions within Li-S batteries more efficiently. Various types of single-atom electrocatalysts have been designed and reported, and the performance of Li-S batteries has been gradually improved. Certainly, single-atom electrocatalysts also have the disadvantages of complicated synthesis methods and high cost. Therefore, how to synthesize single-atom electrocatalysts for Li-S batteries in large quantities in an efficient, fast, and inexpensive manner is a key issue at present. In addition, it can be seen from Table 2 that more electrocatalyst content may affect the specific surface area of the material, which may be related to excessive electrocatalyst agglomeration. At the same time, too much electrocatalyst will also reduce the energy density of Li-S batteries, which is not conducive to the practical application of Li-S batteries. Therefore, it is necessary to ensure the catalytic effect while reasonably controlling the electrocatalyst content.”

Reviewer 2 Report

This work deals with a deep review about the application of nanomaterials in the field of electrocatalysis for Li-S batteries. The manuscript is divided into sections dedicated to homogeneous and heterogeneous catalysts, with a rational and justified classification. The bibliographical analysis is deep and intense, using recent and relevant publications. The conclusions and perspectives are in agreement with the main developments and challenges described during the discussion. The manuscript is of high interest for the journal Nanomaterials and can be published with minimal changes. In order to broaden the joint impact of the article and the journal, the authors could take into account the following suggestion:

In Table 1 and in the discussions of the two sections, the authors could add the following articles published in the journal Nanomaterials: 10.3390/nano12213770; 10.3390/nano12183104; 10.3390/nano12142403; 10.3390/nano10030424 ; 10.3390/nano11081910.

Author Response

This work deals with a deep review about the application of nanomaterials in the field of electrocatalysis for Li-S batteries. The manuscript is divided into sections dedicated to homogeneous and heterogeneous catalysts, with a rational and justified classification. The bibliographical analysis is deep and intense, using recent and relevant publications. The conclusions and perspectives are in agreement with the main developments and challenges described during the discussion. The manuscript is of high interest for the journal Nanomaterials and can be published with minimal changes. In order to broaden the joint impact of the article and the journal, the authors could take into account the following suggestion:

In Table 1 and in the discussions of the two sections, the authors could add the following articles published in the journal Nanomaterials: 10.3390/nano12213770; 10.3390/nano12183104; 10.3390/nano12142403; 10.3390/nano10030424; 10.3390/nano11081910.

RESPONSE: Thank you for your reminder. The references you listed are very valuable and helpful, which are of great significance for enriching this manuscript. We list and cite relevant references published in the journal Nanomaterials in both Table 1 and the text:

Heterogeneous electrocatalyst

Material

Nanostructure

Cycle performance [mA h g−1]

Ref.

Metal compound electrocatalyst

Titanium-deficient anatase TiO2

0D

511/ 2 C/ 1000th

[89]

Fe3O4 nanocrystals/CNTs

0D/1D

538.5/1 C/1800th

[83]

Pt NPs/graphene

0D/2D

789/0.1 C/100th

[57]

CNTs /V2O5 nanoplates

1D/2D

887.2/1 C/500th

[95]

Fe2O3 NPs/ Porous graphene

0D/3D

516/5 C/1000th

[77]

FeS2 nanoclusters/ Porous carbon

0D/3D

1171/0.4 C/100th

[115]

FeP/Carbon nanoarray

1D

500/1 C/200th

[120]

MnO2 nanosheets

2D/

245/2 C/2000th

[86]

Graphene-like Co9S8

2D

643/2 C/400th

[109]

Layered MoS2

2D

over 800/ 0.2 C/300th

[113]

VN nanoribbon/Graphene

3D

917/1 C/200th

[100]

Mesoporous TiN

3D

644/ 0.5 C/500th

[101]

MIL-88A

3D

200/ 0.5 C/1000th

[122]

Nanoporous Co and N-Codoped Carbon Composite

3D

477/ 1 C/1000th

[123]

Heterostructures electrocatalyst

TiO2-TiN

0D/0D

704/1 C /2000th

[128]

CoS2-Co

0D/0D

1227/0.2 C/150th

[129]

MoP2 NPs/CNTs

0D/2D

543/1 C/500th

[138]

TiO2/Ti3C2Tx

0D/2D

576/2 C/1000th

[133]

CoZn-Se NPs/MXene

0D/2D

849/3 C/300th

[148]

TiO2 nanodots/COFs

0D/2D

875/0.5 C/800th

[152]

Nitrogen-doped MXene/nitrogen-doped carbon nanosheet-nickel

0D/2D

588/1 C/500th

[140]

CuCo2S4 NPs/CNTs

0D/2D

627/1 C/500th

[135]

Co NPs/ N-doped graphitic carbon

0D/3D

625/1 C/500th

[147]

COFs/MOFs

2D/3D

636.2/1 C/500th

[153]

WS2-rGO-CNTs

0D/1D/2D (3D)

1227/0.1 C/100th

[139]

CuZnS QDs-decorated (NiCo)–S@rGO/oxdCNT

0D/1D/2D (3D)

537/0.1C/100th

[134]

Single atoms electrocatalyst

SAV/ Graphene

0D/2D

551/0.5 C/400th

[62]

CoSA/Carbon nanosheet

0D/2D

675/1 C/1000th

[171]

Mo-N2/Cabon nanosheet

0D/2D

817/2 C/550th

[180]

SAFe/N-porous carbon

0D/3D

427/0.1 C/300th

[166]

SACo/Hollow carbon spheres

0D/3D

692/1 C/500th

[173]

Co-P cluster/Carbon matrix

0D/3D

746/1 C/1000th

[175]

References

[122] Benítez, A.; Amaro-Gahete, J.; Esquivel, D.; Romero-Salguero, F.J.; Morales, J.; Caballero, Á. MIL-88A metal-organic frame-work as a stable sulfur-host cathode for long-cycle Li-S batteries. Nanomaterials 2020, 10, 424. https://doi.org/10.3390/nano10030424

[123] Niu, S.; Hu, C.; Liu, Y.; Zhao, Y.; Yin, F. Nanoporous Co and N-codoped carbon composite derived from ZIF-67 for high-performance lithium-sulfur batteries. Nanomaterials 2021, 11, 1910. https://doi.org/10.3390/nano11081910

[134] Artchuea, T.; Srikhaow, A.; Sriprachuabwong, C.; Tuantranont, A.; Tang, I.-M.; Pon-On, W. Copper zinc sulfide (CuZnS) quantum dot-decorated (NiCo)–S/conductive carbon matrix as the cathode for Li–S batteries. Nanomaterials 2022, 12, 2403. https://doi.org/10.3390/nano12142403

[135] Wang, H.; Song, Y.; Zhao, Y.; Zhao, Y.; Wang, Z. CuCo2S4 Nanoparticles embedded in carbon nanotube networks as sulfur hosts for high performance lithium-sulfur batteries. Nanomaterials 2022, 12, 3104. https://doi.org/10.3390/nano12183104

[140] Yi, R.; Zhao, Y.; Liu, C.; Sun, Y.; Zhao, C.; Li, Y.; Yang, L.; Zhao, C. A Ti3C2Tx-based composite as separator coating for stable Li-S batteries. Nanomaterials 2022, 12, 3770. https://doi.org/10.3390/nano12213770

Reviewer 3 Report

This manuscript look good in terms of organization and flow. I have the following minor corrections for this manuscript with recommendation to publish. 

The English needs to improve more. 

At the end of the introduction, the major contributions of this review needs to be more hi-lighted as compared to the existing reviews available in this area. 

In the concluding part, the perspective needs to be more improved as per the content of the manuscript. 

Author Response

This manuscript look good in terms of organization and flow. I have the following minor corrections for this manuscript with recommendation to publish.

The English needs to improve more.

RESPONSE: Thanks for your comment. We have made some linguistic changes and additions to the manuscript to make its content more academic and rigorous. The specific modifications are as follows:

“Inspired by this, traditional electrocatalyst Pt nanoparticles (NPs) are loaded on graphene as the cathode host of Li-S batteries (Fig. 3a), and found that its specific capacity was about 40% higher than that of conventional graphene/sulfur composite cathodes (Fig. 3b).” (Lines 153~156)

“The modification of Au NPs on commercial acetylene black/sulfur composite cathodes (CB-S-Au) is also capable of electrocatalysis. Rapid conversion of polysulfides is achieved by regulating the chemistry between Au NPs and polysulfides, thus improving the performance of Li-S batteries (Fig. 3c, d).” (Lines 157~161)

“Loading α-Fe2O3 NPs on three-dimensional (3D) porous graphene (Fig. 4a) as the cathode of Li-S batteries can achieve electrocatalytic effects.” (Lines 179~181)

“The introduction of Fe3O4-encapsulated NPs into 1D carbon nanotubes (CNTs) as hosts for Li-S batteries can effectively accelerate the electrochemical reactions inside the batteries.” (Lines 186~188)

“Zero-dimensional (0D) ultrafine Fe3O4 nanocrystals were introduced in situ into carbon nanospheres as electrocatalysts to modify the separator, which can effectively enhance the electrochemical performance of Li-S batteries.” (Lines 193~196)

“Two-dimensional (2D) ultrathin MnO2 nanosheets were shown to accelerate the conversion of polysulfides to insoluble Li2S/Li2S2.” (Lines 210~211)

“Titanium-deficient anatase TiO2 (TDAT) as a separator modification material can also improve the electrochemical performance of Li-S batteries.” (Lines 220~222)

“Besides TiO2, nano-Magnéli-phase titanium oxide (TinO2n-1) also occupies an important position in the electrocatalysis of Li-S batteries. During the electrochemical reaction, TinO2n-1 is capable of bonding or electrocatalysis with sulfur, which has a significant improvement on the battery performance.” (Lines 226~229)

“The composite of VN nanoribbons with porous graphene and used as cathode materials for Li-S batteries can effectively accelerated the reaction kinetics inside the batteries.” (Lines 225~257)

“Mesoporous TiN with high specific surface area synthesized by solid-solid phase separation method used zinc titanate as raw material exhibited good electrochemical properties.” (Lines 270~272)

“Therefore, the introduction of WN with different nanostructures into various types of carbon materials can exhibit good electrocatalytic properties.” (Lines 277~279)

“A Co9S8 with interconnected graphene-like nanostructures was synthesized and demonstrated to interact strongly with polysulfides when used as a cathode.” (Lines 302~303)

“For example, Mo6S8 can be chelated with polysulfides to form LixMo6S8.” (Lines 311~312)

“Therefore, there is a difference in the properties between the unsaturated-coordinated edge metal sites and the saturated-coordinated surface metal sites in MoS2.” (Lines 333~335)

“In addition, CoS2 was also added as an electrocatalyst to the carbon/sulfur cathode to improve the performance of Li-S batteries by accelerating the redox kinetics of polysulfides.” (Lines 337~339)

“FeS2 is also used as an electrocatalyst in Li-S batteries. Monodisperse sub-10 nm FeS2 nanoclusters were modified on porous carbon surfaces for enhanced electrochemical kinetics.” (Lines 342~344)

“To further explore the potential of metal sulfides for electrocatalysis in Li-S batteries, the ability of various metal sulfides to adsorb polysulfides and oxidize Li2S was systematically explored.” (Lines 345~347)

“In 2018, Co-O-P was found to be formed in surface oxidized CoP, which enables Co sites to interact strongly with polysulfides and generate Co-S bonds.” (Lines 370~372)

“In addition, the special structure of the carbon nanoarray-coated FeP materials can facilitates the fast conduction of ions/electrons.” (Lines 375~376)

“Due to the complex electrocatalytic properties, it is necessary to explore in depth the electrocatalytic effect of various types of transition metal phosphides on Li-S batteries.” (Lines 379~381)

“In response to this problem, the concept of bifunctional heterogeneous structure was proposed. The combination of metal oxides with high adsorption and metal nitrides with high conductivity enables the effective preparation of heterogeneous structured electrocatalysts.” (Lines 414~418)

“Based on this idea, a checkerboard-like heterostructure electrocatalyst was designed by the sulfonation reaction.” (Lines 421~422)

“A Ti3C2Tx/TiO2 heterostructure catalysts was successfully prepared by in situ oxidation and used as interlayers.” (Lines 428~430)

“Aiming at the problem of good adsorption but poor conductivity of VO2, in situ growth of VO2 on the surface of 2D reduced graphene oxide (rGO) by solvothermal method to construct heterostructures is a good solution.” (Lines 451~453)

“Aiming at the problem of good adsorption but poor conductivity of VO2, in situ growth of VO2 on the surface of 2D reduced graphene oxide (rGO) by solvothermal method to construct heterostructures is a good solution.” (Lines 153~156)

“A heterostructure composed of CNTs and MoP2 NPs was combined and applied to the interlayer of Li-S batteries.” (Lines 457~458)

“Carbonization of MOF precursors is a common method to prepare MOF-based hetero-structures. For example, 3D porous cobalt-N-doped graphitic carbon (Co-N-GC) hetero-structures can be obtained by carbonizing Co-MOF.” (Lines 489~492)

“Based on the interaction between MOF and MXene, bimetallic selenides on nitro-gen-doped MXene (CoZn-Se@N-MX) enable successful self-assembly.” (Lines 496~498)

“Homogeneous embedding of anatase/rutile TiO2 nanodots (10 nm) into porous COFs enables to obtain a hybrid-crystal-phase TiO2/COFs heterostructure.” (Lines 519~521)

“In addition, calculations were able to demonstrate that the binding energy of different phases of TiO2 on polysulfides were different.” (Lines 523~524)

“Wrapping the porous MOFs with microporous COFs enables to obtain heterogeneous structures with advanced nanostructures, where the inner MOFs act as catalysts to accelerate the conversion of polysulfides, while the outer COFs layer restricts the shuttle of polysulfides through polar functional groups.” (Lines 528~532)

“For this reason, the interaction between isolated atoms and supports was proposed in 1978, through which the movement and aggregation of isolated atoms can be prevented and a uniform and stable distribution of active sites can be constructed, thus positively affecting the selectivity, stability and electrocatalytic activity of the catalysts.” (Lines 564~568)

“SACs with saturated coordination configurations were first applied to Li-S batteries in 2018, when single-atom Fe (SAFe) coordinated with four pyrrole N on the surface of the supports was used as an electrocatalyst to improve the rate performance and cycling life-time of the battery, demonstrating the electrocatalytic effect of SAFe on the redox reaction of LiPSs. To investigate the intrinsic mechanism of SACs in the electrochemical reaction of Li-S batteries, a novel theoretical simulation study was proposed, which demonstrated that SAFe can coordinate with Li2S thus weakening the Li-S bond, promoting the decomposition of Li2S and the growth of LiPSs chains, and that it can regenerate by gaining electrons during the charging phase (Fig. 15a).” (Lines 578~587)

“Therefore, based on the purpose of screening the most suitable metal atoms for the preparation of SACs for catalytic Li-S electrocatalysis, a novel theoretical model verified the effect of SACs constructed with different monometallic atoms in saturated coordination configurations on the electrocatalytic capacity of Li-S batteries.” (Lines 596~600)

“To deeply dissect the microstructure, the homogeneous dispersion state of Co on the support was analyzed by aberration-corrected scanning transmission electron microscopy (STEM) high-angle annular dark field (HAADF) (Fig. 16c) imaging, x-ray absorption near edge structure (XANES) (Fig. 16d) and extended x-ray absorption fine structure (EXAFS) (Fig. 16e), which elucidate the Co-N4 saturated coordination conformation of such SACs at the atomic level (Fig. 16f).” (Lines 632~638)

“The SACs constructed by co-coordination of N, S atoms also exhibited significant superiorities in accelerating the Li-S electrochemical reactions, for example, single-atom catalysts with Co-N3S-C coordination structures prepared by thiourea (Fig. 17a).” (Lines 648~650)

“With the purpose of simultaneously solving the problems of lithium dendrites and shuttle effects inherent to Li-S batteries, single-atom array mimics with Co-O4 coordination structures constructed from periodically arranged Co and O atoms on the surface of ultrathin metal-organic frameworks were successfully applied to Li-S batteries (Figure 17e).” (Lines 658~661)

“Based on the work of SACs, a dual-atom catalyst Co-P cluster/NC using P atoms as ligand atoms as well as catalytic progenitors (Fig. 18a) was proposed, and the local structure and chemical environment of Co atoms were further analyzed by XANES and WT-EXAFS (Fig. 18b), which demonstrated the Co-P cluster/NC configuration constructed by coordination of atomic Co with P, N.” (Lines 681~685)

“The introduction of highly folded structures into the support can modify the local coordination structure of the FeN4 group (Fig. 18e) thus achieving a simultaneous enhancement of the electrocatalytic activity of SACs at different current densities.” (Lines 689~692)

“To explore the effect of the coordination number of atoms on the catalytic activity of SACs, the investigators demonstrated the stronger adsorption capacity and facilitated conversion kinetics of LiPSs for the oversaturated coordination configuration of Ni-N5 by a com-bination of DFT calculations, electrochemical tests, and spectroscopic characterization.” (Lines 715~719)

“Fe-N5-type SACs in supersaturated coordination configuration demonstrated that Fe monoatoms with high N coordination number can strengthen the immobilization of LiPSs compared to saturated coordination (Fig. 19d) and effectively reduce the energy barriers for the conversion of LiPSs and Li2S nucleation (Fig. 19e, f). In order to address the problem of insufficient adsorption capacity of M-N4 configuration SACs for polysulfides, Fe-N3C2-C SACs constructed by using C and N as co-coordinating atoms were designed and prepared with asymmetric supersaturated coordination configurations, which not only form additional π-bonds with the p orbital hybridization of S to anchor LiPSs, but also effectively enhance the redox kinetics of LiPSs.” (Lines 724~732)

“Single-atom catalysts with Mo-N2-C conformation (Fig. 20a) with good adsorption of LiPSs and Li2S (Fig. 20b) and improved conversion kinetics (Fig. 20c) were applied to obtain Li-S batteries with high reversible capacity (743.9 mAh g-1 at 5C rate) and long cycling lifetime (after 550 cycles with a capacity decay of only 0.018% per cycle). Through a combination of supporting structure design and coordination environment modulation, SACs with FeN2 unsaturated coordination structures were introduced onto graphene with a pore-rich structure (Figure 20d), achieving uniform Li transport and effective adsorption of LiPSs.” (Lines 744~752)

“Nickel dimethoxyethane chloride adduct (NiDME) were selected as homogeneous catalysts for accelerating the redox conversion of sulfur as well as for inhibiting the shuttle effect of polysulfides.” (Lines 844~846)

“Anthraquinone derivative 1,5-bis(2-(2-(2-methoxyethoxy) ethoxy) ethoxy) an-thra-9,10-quinone (AQT) were incorporated into the electrolyte, which effectively enhanced the utilization of Li2S and prevented the deposition of "dead" Li2S.” (Lines 863~865)

“Further, AQT were applied to all-solid-state Li-S batteries, using its fast electron transfer pathways constructed between active materials and current collector to greatly reduce the dissociation barrier of Li2S (Fig. 22g), thus achieving fast Li-S redox reaction kinetics in a solid-state battery.” (Lines 876~879)

“To suppress the loss/deactivation of catalyst, the heterogeneous electrocatalyst CoSNC (CoS1.097 nanoparticles embedded in nitrogen-doped porous carbon sheets) were rationally coupled with the homogeneous electrocatalyst CoCp2 (cobaltocene), and their synergy was utilized to catalyze the electrochemical reaction of polysulfides (Fig. 22h, i).” (Lines 881~885)

“Based on this consideration, a diphenyl diselenide (DPDSe) was introduced into electrolyte, which could spontaneously react with the polysulfide to generate intermediate lithium phenyl selenium polysulfides (LiPhSePSs) with improved redox-mediated ability (Fig. 23a), thus accelerating the polysulfide bidirectional redox kinetics in reaction mechanism.” (Lines 905~909)

“Along the line of changing reaction mechanism, homogeneous electrocatalyst allyl methyl disulfide (AMDS)-mediated all-liquid-phase mechanism were further applied to Li-S batteries over a wide temperature range (room temperature to Cryogenic temperature).” (Lines 928~931)

“Starting from the design described above, a semi-immobilization strategy for the co-polymerization of small molecule imides with medium chain polyether to form space-constrained but catalytically active semi-immobilized RMs (PIPEs) was firstly pro-posed.” (Lines 954~956)

“Based on the semi-immobilization strategy, the semi-immobilized electrocatalyst were further extended to heterogeneous electrocatalysis, which verified the feasibility of coupling homogeneous electrocatalysis with non-homogeneous electrocatalysis.” (Lines 980~982)

At the end of the introduction, the major contributions of this review needs to be more hi-lighted as compared to the existing reviews available in this area.

RESPONSE: Thanks for your kind comment. We have supplemented the manuscript with additional relevant tables giving an overview of heterogeneous electrocatalysts and homogeneous electrocatalysts, respectively, and summarized the advantages and disadvantages of both types of electrocatalysts in order to highlight the main contributions of this manuscript. The specific changes to the introduction section are as follows:

“The nanostructures and functional groups of the two types of electrocatalysts are systematically summarized separately. The relationship between the amount of non-homogeneous electrocatalyst, the specific surface of the material structure and the electrochemical performance is also discussed. Finally, future design ideas are proposed based on the existing achievements, current challenges, and potential opportunities to re-examine the important development of nanostructures and molecular structures in Li-S battery electrocatalysts. In particular, a generic scheme comparing heterogeneous and homogeneous electrocatalyst mechanism is reported in order to better explain the two different process/mechanism. All in all, trying to interpret the electrocatalysis of Li-S batteries from the chemical perspective of structure-determining properties will open up new ideas for the practical large-scale application of Li-S batteries.” (Lines 121~124, 127~129)

In the concluding part, the perspective needs to be more improved as per the content of the manuscript.

RESPONSE: RESPONSE: Thank you for your comment. As your suggestion, we have enriched our content in the conclusion. And according to the content, we have enriched our viewpoint and summarized the advantages and disadvantages of heterogeneous electrocatalysts and homogeneous electrocatalysts, and enriched the relevant content. (Lines 1112~1122)

“We propose a general strategy for comparing the advantages and disadvantages of heterogeneous and homogeneous electrocatalyst to discuss the above issues. We propose a general strategy for comparing the advantages and disadvantages of heterogeneous and homogeneous electrocatalyst to discuss the above issues. Since the two types of electrocatalysis have different characteristics and mechanisms, we believe that selecting an electrocatalyst based on the need for electrochemical performance is an appropriate strategy, specifically, evaluating and selecting an electrocatalyst based on the balance of electrocatalytic performance and catalytic activity retention (Table 4). We hope that this strategy can provide a reference for the subsequent research direction and selection strategy of lithium-sulfur battery electrocatalysts, and provide some help for the research of novel electrocatalysts.”

Table 4. Comparison of the advantages and disadvantages of heterogeneous and homogeneous electrocatalysts and the criteria for judging the process/mechanism of electrocatalysts.

Types of electrocatalysts

Advantages

Disadvantages

Process/Mechanism judgment basis

Heterogeneous

electrocatalyst

1. Convenient and efficient electrocatalytic mechanism

2. Catalytic performance conveniently modulated by nanostructures

3. Improved anode structure and performance

1. Decreased activity as the active surface be gradually covered

2. As an inactive component affects battery energy density

Judgment in the balance between electrocatalytic performance and catalytic activity retention based on performance requirements

(Heterogeneous

electrocatalysts usually provide more direct electrochemical performance enhancement but less stability, while homogeneous electrocatalysts provide more stable electrocatalytic activity)

Homogeneous electrocatalyst

1. Enhanced kinetics in terms of reaction mechanism

2. Active site enrichment

(Full exposure to active substances)

1. Decreased activity with shuttle effect

2. Complex catalytic mechanism

Reviewer 4 Report

In this work the authors provided a review of Advanced nanostructured materials for electrocatalysis in lithium-sulfur batteries. A classification of different types of nanomaterials and catalytic mechanism, such as heterogeneous electrocatalytic and homogeneous electrocatalytic systems was provided and explained. The manuscript contains interesting information and fits within the aims and scope of Nanomaterials. The manuscript is clearly written. Thus, I recommend accepting this paper after a minor revision.

Belows are some suggestions:

In general some recent work should be cited in the manuscript to update the bibliography.

Page 2 line 85: The authors mentioned different strategies of physical and chemical confinement of LiPSs - "constructing cathodes with novel nanostructures to confine polysulfides; modifying separators to suppress the shuttle effect; rationally designing novel electrolytes to confine polysulfide shuttles; and constructing robust artificial solid electrolyte interfaces on Li anodes to suppress lithium dendrites."- Other strategies should be mentioned as for example double-layer/interlayer directly coated on the electrode surface approach to confine LiPSs, implementing in the interlayer catalytic materials (Applied Materials Today 25 (2021) 101169 – doi: 10.1016/j.apmt.2021.101169, Journal of Power Sources 521 (2022) 230945 – doi: 10.1016/j.jpowsour.2021.230945)

Page 6 line 217: other Titanium-based nanomaterials have to be mentioned, such as nano-Magnéli-phase titanium oxide (TinO2n-1)

Paragraph 2.1.2., page 7 line 229: the catalytic role of nanoscaled vanadium nitride oxide (VOxNy) have to be mentioned and/or explained

Paragraph 2.1.2., Page 9 line 278: electrocatalysis of polysulfide conversion such as defected metal sulfides such as MoS2-x should be reported (such as Energy Environ. Sci., 2017, 10, 1476-1486 – doi: 10.1039/C7EE01047H)

Page 14 line 387: The authors reported "The synergistic effect of multi-component materials to exert different functions can further improve the performance of Li-S batteries" the bibliography should be improved and enriched, an example could be MoS2 combined with PANI.

Paragraph 2.2.2. page 15 line 421: 2D materials with large accessible surface such as nitrogen doped carbon materials or carbon nitride should be included

Page 17 line 497: some typo issues should be corrected "rutile-TiO2"

A table comparing the different nanomaterial, families of homogeneous electrocatalysts and the relative performances should be reported in order to better compare and highlight the role of the catalysts, similarly as reported for Heterogeneous electrocatalyst in table 1.

A generic scheme comparing Heterogeneous and homogeneous electrocatalyst mechanism should be reported in the manuscript in order to better explained the two different process/mechanism.

Author Response

In this work the authors provided a review of Advanced nanostructured materials for electrocatalysis in lithium-sulfur batteries. A classification of different types of nanomaterials and catalytic mechanism, such as heterogeneous electrocatalytic and homogeneous electrocatalytic systems was provided and explained. The manuscript contains interesting information and fits within the aims and scope of Nanomaterials. The manuscript is clearly written. Thus, I recommend accepting this paper after a minor revision.

Belows are some suggestions:

In general, some recent work should be cited in the manuscript to update the bibliography.

RESPONSE: Thanks for your kind comment. Your suggestions are important to enrich this manuscript. We have cited additional relevant references from the last three years (2020~2022) in all sections of the manuscript.

References

[43] Versaci, D.; Cozzarin, M.; Amici, J.; Francia, C.; Leiva, E. P.M.; Visintin, A.; Bodoardo, S. Influence of synthesis parameters on g-C3N4 polysulfides trapping: A systematic study. Appl. Mater. Today 2021, 25, 101169. https://doi.org/10.1016/j.apmt.2021.101169

[44] Versaci, D.; Canale, I.; Goswami, S.; Amici, J.; Francia, C.; Fortunato, E.; Martins, R.; Pereira, L.; Bodoardo, S. Molybdenum disulfide/polyaniline interlayer for lithium polysulphide trapping in lithium-sulphur batteries. J. Power Sources 2022, 521, 230945. https://doi.org/10.1016/j.jpowsour.2021.230945

[90] Zhang, H.; Yang, L.; Zhang, P. G.; Lu, C. J.; Sha, D. W.; Yan, B. Z.; He, W.; Zhou, M.; Zhang, W.; Pan, L.; Sun, Z. M. MXene-derived TinO2n−1 quantum dots distributed on porous carbon nanosheets for stable and long-life Li–S batteries: Enhanced polysulfide mediation via defect engineering. Adv. Mater. 2021, 33, 2008447. https://doi.org/10.1002/adma.202008447

[91] Xia, J.; Gao, R. H.; Yang, Y.; Tao, Z.; Han, Z. Y.; Zhang, S. C.; Xing, Y. L.; Yang, P. H.; Lu, X.; and Zhou G. M. TinO2n–1/MXene hierarchical bifunctional catalyst anchored on graphene aerogel toward flexible and high-energy Li–S batteries. ACS Nano 2022, 16, 19133-19144. https://doi.org/10.1021/acsnano.2c08246

[103] Zubair, U.; Bianco, S.; Amici, J.; Francia, C.; Bodoardo, S. Probing the interaction mechanism of heterostructured VOxNy nanoparticles supported in nitrogen-doped reduced graphene oxide aerogel as an efficient polysulfide electrocatalyst for stable sulfur cathodes. J. Power Sources 2020, 461, 228144. https://doi.org/10.1016/j.jpowsour.2020.228144

[122] Benítez, A.; Amaro-Gahete, J.; Esquivel, D.; Romero-Salguero, F.J.; Morales, J.; Caballero, Á. MIL-88A metal-organic frame-work as a stable sulfur-host cathode for long-cycle Li-S batteries. Nanomaterials 2020, 10, 424. https://doi.org/10.3390/nano10030424

[123] Niu, S.; Hu, C.; Liu, Y.; Zhao, Y.; Yin, F. Nanoporous Co and N-codoped carbon composite derived from ZIF-67 for high-performance lithium-sulfur batteries. Nanomaterials 2021, 11, 1910. https://doi.org/10.3390/nano11081910

[127] Shi, M. J.; Liu, Z.; Zhang, S.; Liang, S. C.; Jiang, Y. T.; Bai, H., Jiang, Z. M.; Chang, J.; Feng, J.; Chen, W. S.; Yu, H. P.; Liu, S.X.; Wei, T.; Fan, Z. J. A mott–schottky heterogeneous layer for Li–S Batteries: Enabling both high stability and commercial-sulfur utilization. Adv. Energy Mater. 2022, 12, 2103657. https://doi.org/10.1002/aenm.202103657

[134] Artchuea, T.; Srikhaow, A.; Sriprachuabwong, C.; Tuantranont, A.; Tang, I.-M.; Pon-On, W. Copper zinc sulfide (CuZnS) quantum dot-decorated (NiCo)–S/conductive carbon matrix as the cathode for Li–S batteries. Nanomaterials 2022, 12, 2403. https://doi.org/10.3390/nano12142403

[135] Wang, H.; Song, Y.; Zhao, Y.; Zhao, Y.; Wang, Z. CuCo2S4 Nanoparticles embedded in carbon nanotube networks as sulfur hosts for high performance lithium-sulfur batteries. Nanomaterials 2022, 12, 3104. https://doi.org/10.3390/nano12183104

[140] Yi, R.; Zhao, Y.; Liu, C.; Sun, Y.; Zhao, C.; Li, Y.; Yang, L.; Zhao, C. A Ti3C2Tx-based composite as separator coating for stable Li-S batteries. Nanomaterials 2022, 12, 3770. https://doi.org/10.3390/nano12213770

[141] Zhang, T. P.; Hu, F. Y.; Song, C.; Li, S. M.; Shao, W. L.; Liu, S. Y.; Peng, H.; Hu, S.; Jian, X. G. Constructing covalent triazine-based frameworks to explore the effect of heteroatoms and pore structure on electrochemical performance in Li–S batteries. Chem. Eng. J. 2021, 407, 127141. https://doi.org/10.1016/j.cej.2020.127141

[142] Zhang, C. Q.; Du, R. F.; Biendicho, J. J.; Yi, M. J.; Xiao, K.; Yang, D. W.; Zhang, T.; Wang, X.; Arbiol, J.; Llorca, J.; Zhou, Y. T.; Morante, J. R.; Cabot, A. Tubular CoFeP@CN as a mott–schottky catalyst with multiple adsorption sites for robust lithium−sulfur batteries. Adv. Energy Mater. 2021, 11, 2100432. https://doi.org/10.1002/aenm.202100432

[143] Xiao, Q. H. Q.; Li, G. R.; Li, M. J.; Liu, R. P.; Li, H. B.; Ren, P. F.; Dong, Y.; Feng, M.; Chen, Z. W. Biomass-derived nitrogen-doped hierarchical porous carbon as efficient sulfur host for lithium–sulfur batteries. J. Energy Chem. 2020, 44, 61-67. https://doi.org/10.1016/j.jechem.2019.09.004

Page 2 line 85: The authors mentioned different strategies of physical and chemical confinement of LiPSs - "constructing cathodes with novel nanostructures to confine polysulfides; modifying separators to suppress the shuttle effect; rationally designing novel electrolytes to confine polysulfide shuttles; and constructing robust artificial solid electrolyte interfaces on Li anodes to suppress lithium dendrites."- Other strategies should be mentioned as for example double-layer/interlayer directly coated on the electrode surface approach to confine LiPSs, implementing in the interlayer catalytic materials (Applied Materials Today 25 (2021) 101169 – doi: 10.1016/j.apmt.2021.101169, Journal of Power Sources 521 (2022) 230945 – doi: 10.1016/j.jpowsour.2021.230945)

RESPONSE: Thank you for your comment. As your suggestion, we add to these strategies the design of the additional interlayer materials and cite some of the relevant references in the manuscript. (Lines 87~88)

“These strategies mainly include: constructing cathodes with novel nanostructures to con-fine polysulfides; modifying separators to suppress the shuttle effect; introduction of additional interlayer materials with special properties to anchor polysulfides [43,44]; ration-ally designing novel electrolytes to confine polysulfide shuttles; and constructing robust artificial solid electrolyte interfaces on Li anodes to suppress lithium dendrites.”

References

[43] Versaci, D.; Cozzarin, M.; Amici, J.; Francia, C.; Leiva, E. P.M.; Visintin, A.; Bodoardo, S. Influence of synthesis parameters on g-C3N4 polysulfides trapping: A systematic study. Appl. Mater. Today 2021, 25, 101169. https://doi.org/10.1016/j.apmt.2021.101169

[44] Versaci, D.; Canale, I.; Goswami, S.; Amici, J.; Francia, C.; Fortunato, E.; Martins, R.; Pereira, L.; Bodoardo, S. Molybdenum disulfide/polyaniline interlayer for lithium polysulphide trapping in lithium-sulphur batteries. J. Power Sources 2022, 521, 230945. https://doi.org/10.1016/j.jpowsour.2021.230945

Page 6 line 217: other Titanium-based nanomaterials have to be mentioned, such as nano-Magnéli-phase titanium oxide (TinO2n-1)

RESPONSE: Thank you for your comment. As your suggestion, it is not enough to introduce only TiO2 about titanium-based materials, other titanium-based materials should be mentioned as well. We have added a description of the application of TinO2n-1 in lithium-sulfur battery electrocatalysis and cited references. (Lines 226~229)

“Besides TiO2, nano-Magnéli-phase titanium oxide (TinO2n-1) also occupies an important position in the electrocatalysis of Li-S batteries. During the electrochemical reaction, TinO2n-1 is capable of bonding or electrocatalysis with sulfur, which has a significant im-provement on the battery performance [90, 91].”

References

[90] Zhang, H.; Yang, L.; Zhang, P. G.; Lu, C. J.; Sha, D. W.; Yan, B. Z.; He, W.; Zhou, M.; Zhang, W.; Pan, L.; Sun, Z. M. MXene-derived TinO2n−1 quantum dots distributed on porous carbon nanosheets for stable and long-life Li–S batteries: Enhanced polysulfide mediation via defect engineering. Adv. Mater. 2021, 33, 2008447. https://doi.org/10.1002/adma.202008447

[91] Xia, J.; Gao, R. H.; Yang, Y.; Tao, Z.; Han, Z. Y.; Zhang, S. C.; Xing, Y. L.; Yang, P. H.; Lu, X.; and Zhou G. M. TinO2n–1/MXene hierarchical bifunctional catalyst anchored on graphene aerogel toward flexible and high-energy Li–S batteries. ACS Nano 2022, 16, 19133-19144. https://doi.org/10.1021/acsnano.2c08246

Paragraph 2.1.2., page 7 line 229: the catalytic role of nanoscaled vanadium nitride oxide (VOxNy) have to be mentioned and/or explained

RESPONSE: Thank you for your comment. As your suggestion, we supplement the section on Metal Nitrides with an introduction to nanoscaled vanadium nitride oxide VOxNy and cite relevant references to illustrate its special electrocatalytic mechanism. (Lines 260~265)

“In addition to VN, vanadium nitride oxides (VOxNy) have also received a lot of attention recently [103]. Unlike other electrocatalytic materials, VOxNy exhibit a redox potential window between their oxide counterparts, around which polysulfides are able to form polysulfide complexes and enhance kinetics and polysulfide anchoring through the V-N and V-O interfaces.”

References

[103] Zubair, U.; Bianco, S.; Amici, J.; Francia, C.; Bodoardo, S. Probing the interaction mechanism of heterostructured VOxNy nanoparticles supported in nitrogen-doped reduced graphene oxide aerogel as an efficient polysulfide electrocatalyst for stable sulfur cathodes. J. Power Sources 2020, 461, 228144. https://doi.org/10.1016/j.jpowsour.2020.228144

Paragraph 2.1.2., Page 9 line 278: electrocatalysis of polysulfide conversion such as defected metal sulfides such as MoS2-x should be reported (such as Energy Environ. Sci., 2017, 10, 1476-1486 – doi: 10.1039/C7EE01047H)

RESPONSE: Thank you for your comment. Your suggestions will be of great help in improving this manuscript. In addition to MoS2, defective metal sulfides also have a wide range of applications in electrocatalysis. As your suggestion, we present the contribution of defective molybdenum-based sulfides (MoS2-x) to polysulfides and cite the relevant references. (Lines 316~319)

“Notably, MoS2 with sulfur deficiencies (MoS2-x) has received much attention due to its high electrochemical activity associated with the presence of sulfur deficiency [111]. Compared with MoS2, MoS2-x can participate in the reaction with polysulfides more effectively and greatly enhance the conversion kinetics of polysulfides [112].”

References

[111] Lin, H. B.; Yang, L. Q.; Jiang, X.; Li, G. C.; Zhang, T. R.; Yao, Q. F.; Zheng, G. Y. W.; Lee, J. Y. Electrocatalysis of polysulfide conversion by sulfur-deficient MoS2 nanoflakes for lithium–sulfur batteries. Energy Environ. Sci. 2017,10, 1476-1486. http://dx.doi.org/10.1039/C7EE01047H

[112] Wang, H. E.; Li, X. C.; Qin, N.; Zhao, X.; Cheng, H.; Cao, G. Z.; Zhang, W. J. Sulfur-deficient MoS2 grown inside hollow mesoporous carbon as a functional polysulfide mediator. J. Mater. Chem. A 2019, 7, 12068-12074. http://dx.doi.org/10.1039/C9TA01722D

Page 14 line 387: The authors reported "The synergistic effect of multi-component materials to exert different functions can further improve the performance of Li-S batteries" the bibliography should be improved and enriched, an example could be MoS2 combined with PANI.

RESPONSE: Thank you for your comment. Your suggestions will enable the reader to better understand this manuscript. We have added additional relevant examples to introduce the compounding between polymers (PANI) and metal compounds. (Lines 408~410)

“It is worth noting that the multi-component materials here include not only the metal compounds mentioned above, but also conductive polymers and other materials [44, 127].”

References

[44] Versaci, D.; Canale, I.; Goswami, S.; Amici, J.; Francia, C.; Fortunato, E.; Martins, R.; Pereira, L.; Bodoardo, S. Molybdenum disulfide/polyaniline interlayer for lithium polysulphide trapping in lithium-sulphur batteries. J. Power Sources 2022, 521, 230945. https://doi.org/10.1016/j.jpowsour.2021.230945

[127] Shi, M. J.; Liu, Z.; Zhang, S.; Liang, S. C.; Jiang, Y. T.; Bai, H., Jiang, Z. M.; Chang, J.; Feng, J.; Chen, W. S.; Yu, H. P.; Liu, S.X.; Wei, T.; Fan, Z. J. A mott–schottky heterogeneous layer for Li–S Batteries: Enabling both high stability and commercial-sulfur utilization. Adv. Energy Mater. 2022, 12, 2103657. https://doi.org/10.1002/aenm.202103657

Paragraph 2.2.2. page 15 line 421: 2D materials with large accessible surface such as nitrogen doped carbon materials or carbon nitride should be included

RESPONSE: Thank you for your comment. Your suggestions are important to enrich the content of this manuscript. We present the application of heteroatom-doped carbon materials in the electrocatalysis of lithium-sulfur batteries. As your suggestion, we additionally introduce electrocatalytic applications of carbon nitride and nitrogen-doped carbon materials (Lines 466~473)

“In addition to the above-mentioned carbon materials with different nanostructures, carbon materials containing heteroatoms are also able to adsorb polysulfides and accelerate their conversion due to their polar functional groups [140,141]. Carbon materials containing N, O, P, S tend to exhibit stronger adsorption-catalysis effects than pure carbon materials. Taking nitrogen-doped carbon materials as an example, carbon nitride [142] or carbon materials containing nitrogen [143] can effectively anchor polysulfide to nitro-gen-containing active sites, thus greatly improving electrochemical properties.”

References

[140] Yi, R.; Zhao, Y.; Liu, C.; Sun, Y.; Zhao, C.; Li, Y.; Yang, L.; Zhao, C. A Ti3C2Tx-based composite as separator coating for stable Li-S batteries. Nanomaterials 2022, 12, 3770. https://doi.org/10.3390/nano12213770

[141] Zhang, T. P.; Hu, F. Y.; Song, C.; Li, S. M.; Shao, W. L.; Liu, S. Y.; Peng, H.; Hu, S.; Jian, X. G. Constructing covalent triazine-based frameworks to explore the effect of heteroatoms and pore structure on electrochemical performance in Li–S batteries. Chem. Eng. J. 2021, 407, 127141. https://doi.org/10.1016/j.cej.2020.127141

[142] Zhang, C. Q.; Du, R. F.; Biendicho, J. J.; Yi, M. J.; Xiao, K.; Yang, D. W.; Zhang, T.; Wang, X.; Arbiol, J.; Llorca, J.; Zhou, Y. T.; Morante, J. R.; Cabot, A. Tubular CoFeP@CN as a mott–schottky catalyst with multiple adsorption sites for robust lithium−sulfur batteries. Adv. Energy Mater. 2021, 11, 2100432. https://doi.org/10.1002/aenm.202100432

[143] Xiao, Q. H. Q.; Li, G. R.; Li, M. J.; Liu, R. P.; Li, H. B.; Ren, P. F.; Dong, Y.; Feng, M.; Chen, Z. W. Biomass-derived nitrogen-doped hierarchical porous carbon as efficient sulfur host for lithium–sulfur batteries. J. Energy Chem. 2020, 44, 61-67. https://doi.org/10.1016/j.jechem.2019.09.004

Page 17 line 497: some typo issues should be corrected "rutile-TiO2"

RESPONSE: Thank you for your comment. We have checked the full manuscript and fixed errors such as misuse of upper and lower case.

A table comparing the different nanomaterial, families of homogeneous electrocatalysts and the relative performances should be reported in order to better compare and highlight the role of the catalysts, similarly as reported for Heterogeneous electrocatalyst in table 1.

RESPONSE: Thank you for your comment. Your suggestions are important to enrich the content of this manuscript. We selected representative homogeneous electrocatalyst references, introduced functional groups, electrical properties and other relevant parameters and plotted tables (Table 3). (Lines 1046~1049)

“Based on recent studies, a brief overview is provided in Table 3, which presents the positive effects of different functional groups/functional structures on the electrocatalysis of Li-S batteries. Hence, a small number of suitable catalysts can effectively optimize the redox kinetics of intrinsic LiPSs and improve the utilization of active materials.”

Table 3. A summary of homogeneous electrocatalyst for Li-S batteries with different functional groups/ structures.

Homogeneous electrocatalyst

Material

Functional group/ structure

Cycle performance [mA h g−1]

Ref.

Homogeneous type electrocatalyst

nickel dimethoxyethane chloride adduct

NiCl2

784/1 C/500th

[66]

Anthraquinone derivative 1,5-bis(2-(2-(2-methoxyethoxy) ethoxy) ethoxy) an-thra-9,10-quinone

Anthraquinone

850/1 C/500th

[182]

CoSNC (CoS1.097 nanoparticles (NPs) embedded in nitrogen-doped porous carbon sheets) with cobaltocene

transition metal metallocenes

509/2 C/1200th

[184]

diphenyl diselenide (DPDSe)

phenylselenides

720/0.5 C/350th

[185]

Allyl methyl disulfide(AMDS)

disulfide structure

798/0.8 C/50th

[186]

Ethyl viologen diperchlorate (EV(ClO4)2)

Ethyl viologen structure

395/1.0 mA cm−2 /500th

[191]

Semi-immobilized type electrocatalyst

PIPE

imide structure

916/0.5 C/300th

[187]

G@ppy-por

porphyrin structure

904/2 C/50th

[188]

BPI

imide structure

560/0.5 C/100th

[189]

CoCp2

transition metal metallocenes

553/2 C/100th

[190]

A generic scheme comparing Heterogeneous and homogeneous electrocatalyst mechanism should be reported in the manuscript in order to better explained the two different process/mechanism.

RESPONSE: Thank you for your comment. Your suggestions are important to enrich the content of this manuscript. As your suggestion, we have enriched our viewpoint and summarized the advantages and disadvantages of heterogeneous electrocatalysts and homogeneous electrocatalysts, and enriched the relevant content. (Lines 1112~1122)

“We propose a general strategy for comparing the advantages and disadvantages of heterogeneous and homogeneous electrocatalyst to discuss the above issues. We propose a general strategy for comparing the advantages and disadvantages of heterogeneous and homogeneous electrocatalyst to discuss the above issues. Since the two types of electrocatalysis have different characteristics and mechanisms, we believe that selecting an electrocatalyst based on the need for electrochemical performance is an appropriate strategy, specifically, evaluating and selecting an electrocatalyst based on the balance of electrocatalytic performance and catalytic activity retention (Table 4). We hope that this strategy can provide a reference for the subsequent research direction and selection strategy of lithium-sulfur battery electrocatalysts, and provide some help for the research of novel electrocatalysts.”

Table 4. Comparison of the advantages and disadvantages of heterogeneous and homogeneous electrocatalysts and the criteria for judging the process/mechanism of electrocatalysts.

Types of electrocatalysts

Advantages

Disadvantages

Process/Mechanism judgment basis

Heterogeneous

electrocatalyst

1. Convenient and efficient electrocatalytic mechanism

2. Catalytic performance conveniently modulated by nanostructures

3. Improved anode structure and performance

1. Decreased activity as the active surface be gradually covered

2. As an inactive component affects battery energy density

Judgment in the balance between electrocatalytic performance and catalytic activity retention based on performance requirements

(Heterogeneous

electrocatalysts usually provide more direct electrochemical performance enhancement but less stability, while homogeneous electrocatalysts provide more stable electrocatalytic activity)

Homogeneous electrocatalyst

1. Enhanced kinetics in terms of reaction mechanism

2. Active site enrichment

(Full exposure to active substances)

1. Decreased activity with shuttle effect

2. Complex catalytic mechanism
